# Identification of metabolic vulnerabilities of receptor tyrosine kinases-driven cancer

Nan Jin[1,2,8], Aiwei Bi[1,2,8], Xiaojing Lan[1,8], Jun Xu[1,2], Xiaomin Wang[1,2], Yingluo Liu[1,2], Ting Wang[1,2], Shuai Tang[1], Hanlin Zeng[1,2], Ziqi Chen[1,2], Minjia Tan [3,2], Jing Ai[1,2], Hua Xie[1,2], Tao Zhang[1,2], Dandan Liu[2,4], Ruimin Huang[2,4], Yue Song[5], Elaine Lai-Han Leung[6], Xiaojun Yao[6], Jian Ding[1,2], Meiyu Geng[1,2], Shu-Hai Lin[7] & Min Huang [1,2]

One of the biggest hurdles for the development of metabolism-targeted therapies is to identify the responsive tumor subsets. However, the metabolic vulnerabilities for most human cancers remain unclear. Establishing the link between metabolic signatures and the oncogenic alterations of receptor tyrosine kinases (RTK), the most well-defined cancer genotypes, may precisely direct metabolic intervention to a broad patient population. By integrating metabolomics and transcriptomics, we herein show that oncogenic RTK activation causes distinct metabolic preference. Specifically, EGFR activation branches glycolysis to the serine synthesis for nucleotide biosynthesis and redox homeostasis, whereas FGFR activation recycles lactate to fuel oxidative phosphorylation for energy generation. Genetic alterations of *EGFR* and *FGFR* stratify the responsive tumors to pharmacological inhibitors that target serine synthesis and lactate fluxes, respectively. Together, this study provides the molecular link between cancer genotypes and metabolic dependency, providing basis for patient stratification in metabolism-targeted therapies.

---

[1] Division of Antitumor Pharmacology, State Key Laboratory of Drug Research, Shanghai Institute of Materia Medica, Chinese Academy of Sciences, 555 Zuchongzhi Road, 201203 Shanghai, China. [2] University of Chinese Academy of Sciences, No.19 Yuquan Road, 100049 Beijing, China. [3] Chemical Proteomics Center, State Key Laboratory of Drug Research, Shanghai Institute of Materia Medica, Chinese Academy of Sciences, 555 Zuchongzhi Road, 201203 Shanghai, China. [4] Shanghai Institute of Materia Medica, Chinese Academy of Sciences, 555 Zuchongzhi Road, 201203 Shanghai, China. [5] Agilent Technologies (China) Co., Ltd., 1350 North Sichuan Road, 200080 Shanghai, China. [6] Macau Institute for Applied Research in Medicine and Health, State Key Laboratory of Quality Research in Chinese Medicine, Macau University of Science and Technology, Avenida Wai Long, Taipa, 999078 Macau, China. [7] State Key Laboratory of Cellular Stress Biology, Innovation Center for Cell Signaling Network, School of Life Sciences, Xiamen University, 4221 South Xiang'an Road, 361102 Xiamen, China. [8] These authors contributed equally: Nan Jin, Aiwei Bi, Xiaojing Lan. Correspondence and requests for materials should be addressed to M.G. (email: mygeng@simm.ac.cn) or to S.-H.L. (email: shuhai@xmu.edu.cn) or to M.H. (email: mhuang@simm.ac.cn)

Metabolic reprogramming, a hallmark of cancer, presents the new therapeutic opportunities and attracts increasing efforts in anticancer drug discovery[1]. Recently, it has been increasingly recognized that biochemical pathways rewiring often induces cancer cell-specific metabolic vulnerabilities. The heterogeneous dependency on nutrient types, also known as metabolic heterogeneity, exists across tumor types[2–6] and even within the same tumor tissues[7–10]. This may largely explain the very limited benefits obtained in the clinical modalities of metabolic targets, in which metabolic inhibitors were often delivered to broad cancer patients without indication of metabolic dependency.

Metabolic reprogramming is believed to result from genetic alterations of metabolic enzymes or putative oncogene activation[11]. At present, only a handful of autonomous abnormalities of metabolic enzyme genes have been identified, such as mutations in isocitrate dehydrogenase (IDH) and amplification of phosphoglycerate dehydrogenase (PHGDH), which were considered as genetic signatures for reprogrammed metabolic network[12–15]. In most cases, a variety of oncogenes orchestrate metabolic reprogramming by inducing broad changes in gene expressions. For instance, oncogenic KRAS and MYC instigate multiple metabolic changes including increasing glucose uptake, differential channeling of glucose metabolism and reprogramming glutathione biosynthesis in cancer[4,16]. Among the plethora of metabolic changes, whether cancer cells with certain oncogenic activation will exhibit a defined metabolic preference is mostly unclear.

Tyrosine kinase receptors (RTK), such as epidermal growth factor receptor (EGFR) and fibroblast growth factor receptor (FGFR), are well recognized oncogenic drivers for the malignant growth of various types of human cancer. To date, RTK gene alterations represent the most well-defined genetic subtypes of human cancers, in particular in non-small cell lung cancer (NSCLC). While mounting evidence has proved the impact of RTK on the rewiring of metabolic network, whether RTK-driven metabolic programming results in metabolic vulnerability with therapeutic potential is still obscure. This study aims to connect the metabolic vulnerability to RTK genotypes, which may provide a feasible approach for patient stratification in metabolism-targeted therapies.

In this study, we show that oncogenic RTK activation causes distinct metabolic preference. EGFR activation branches glycolysis to the serine synthesis for nucleotide biosynthesis and redox homeostasis, whereas FGFR activation recycles lactate to fuel oxidative phosphorylation for energy generation. Our findings provide basis for stratifying EGFR and FGFR aberrant patients for metabolism-targeted therapies.

## Results

**Oncogenic RTK differentially reprogram metabolic phenotypes.** To get a glimpse of the metabolic vulnerabilities of RTK aberrant cancer, we took an approach of pharmacological inhibitor screen. In total 15 NSCLC cell lines covering the high-incidence gene abnormalities, including EGFR mutation (L858R, exon 19 deletion, or exon 21 deletion), FGFR1 amplification, KRAS mutation etc., were exposed to small molecule inhibitors targeting enzymes in glucose and glutamine metabolism or fatty acid oxidation (Supplementary Fig. 1a)[17]. Hierarchical cluster analysis of the growth inhibition rate showed that cancer cells in the same genotype tended to present similar metabolic vulnerabilities, especially for FGFR- and EGFR-aberrant cells that showed a trend of clustering (Supplementary Fig. 1a, Dataset 1). To confirm the clinical relevance of this finding, we extracted 740 lung adenocarcinoma from TCGA database, among which 54

patients were confirmed with EGFR activating mutation ($n = 25$), FGFR1/2 amplification ($n = 15$), MET amplification ($n = 12$), or RET fusion ($n = 2$). In these samples, hierarchical clustering based on the expression of 1498 metabolic genes annotated in KEGG database showed the distinct expression pattern between EGFR-, FGFR- and RET-activated tumors (Supplementary Fig. 1b), suggesting the distinct metabolic phenotypes in oncogenic RTK-driven cancer.

To understand how individual RTK preferentially rewires the metabolic network, we took the advantage of a widely-used BAF3 isogeneic cell model[18–20]. The introduction of the well-validated oncogenic form of EGFR (EGFR-L858R-T790M), FGFR1 (TEL-FGFR1 fusion), MET (TPR-MET fusion) or RET (CCDC6-RET fusion) into BAF3 cells resulted in the constitutively activated RTK signaling (Fig. 1a, Supplementary Fig. 1c), the IL3-independent cell growth (Fig. 1b), and the exquisite sensitivity to specific RTK inhibitors (Fig. 1c). We then characterized the metabolic profiles of these cell lines. It was noted that RTK activation resulted in the enhancement of both aerobic glycolysis and oxidative phosphorylation, as indicated by the extracellular acidification rate (ECAR) and oxygen consumption rate (OCR), but with striking difference between RTK genotypes (Fig. 1d). Given that FGFR gene has four isoforms, we also introduced TEL-FGFR3 fusion into BAF3 cells, which resulted in IL3-independent cell growth (Supplementary Fig. 1d) and the sensitivity to AZD4547 (Supplementary Fig. 1e). The comparison of the FGFR1- and FGFR3-driven BAF3 cells in parallel observed the equally enhanced ECAR and OCR (Supplementary Fig. 1f). We also tested the impact of IL3 on the metabolic phenotypes in these cells, as IL3 is very important for BAF3 cell model. As expected, deprivation of IL3 resulted in the striking change in OCR in BAF3 parental cells, since the survival of these cells is highly dependent on IL3. BAF3-RTK cells were generally much less affected (Supplementary Fig. 1g). The metabolic effect appeared to correlate with the impact of IL3 on cell growth (Fig. 1b).

Further, we performed non-targeted metabolomics in these cell lines using mass spectrometry, which identified 124 metabolites (Supplementary Dataset 2) in distinct metabolic profiling, shown by the heatmap of the individual metabolite abundance (Fig. 1e), and the principal component analysis (PCA) (Fig. 1f). Pathway enrichment analysis of altered metabolites in BAF3-RTK cells (1.5-fold cutoff in relative to parental BAF3 cells; $p < 0.01$) highlighted several metabolic pathways, in particular, citrate cycle (TCA cycle), nucleotide biosynthesis and amino acid metabolism that are required for malignant cell growth. Specifically, TCA cycle was preferentially activated in FGFR-activated cells and glutamine/glutamate metabolism pathway was particularly enhanced in BAF3-RET cells (Supplementary Fig. 1h, Dataset 3).

The heterogeneous metabolic phenotypes may suggest the difference in nutrient acquisition of proliferating cells. As such, we applied the isotopologue spectral analysis (ISA) to examine the incorporation of [U-$^{13}C_6$]-glucose, [U-$^{13}C_5$]-glutamine or [U-$^{13}C_{16}$]-palmitate in the intermediate metabolites. This allowed us to track the metabolism of glucose, glutamine and fatty acids, three major nutrient sources in these cells (Fig. 1g, Supplementary Fig. 1i-k). Heatmap representing the difference of $^{13}$C-labeled metabolites in each cell line highlighted the promoted glucose flux in both BAF3-EGFR and BAF3-FGFR1 cells and the remarkably enhanced glutaminolysis in BAF3-RET cells. Instead, MET amplified cells did not show clear metabolic signature (Fig. 1h, Supplementary Dataset 4).

We then asked whether the metabolic changes in RTK-driven cells could suggest their distinct metabolic dependency. Indeed, we discovered that the proliferation of BAF3-EGFR and BAF3-FGFR1 cells was heavily dependent on glucose supply, whereas

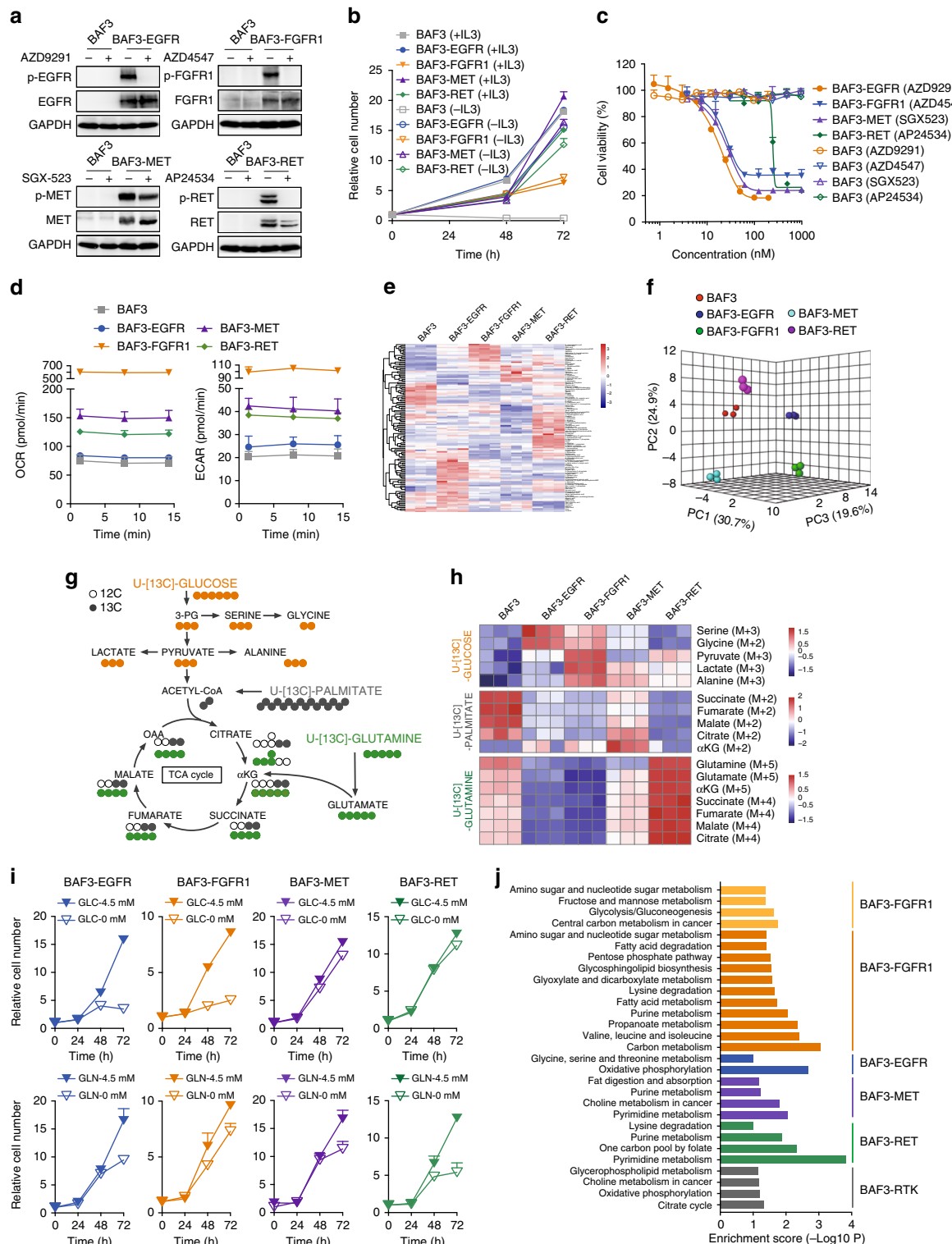

the growth of BAF3-RET cells appeared relying on the glutamine supply (Fig. 1i). These findings were further confirmed in a panel of cancer cell lines bearing similar genetic alterations. 9 *EGFR* mutant cells, 4 *FGFR1/3* amplified cells and 2 *RET* fusion/mutant cells were tested for their growth dependence on glucose or glutamine. 7 wildtype lung cancer cells (except A431) without detectable driving gene alterations annotated by cancer cell line encyclopedia (CCLE) database, were used as control. The details of the genetic alterations in these cells were summarized in Supplementary Dataset 5. Consistent with the observation in

BAF3-RTK cells, the growth of EGFR- and FGFR- activated cells were dependent on glucose rather than glutamine. In contrast, wildtype cells, which lacked driving gene alterations, showed variable growth dependency on glucose or glutamine. *RET* fusion/mutant cells seemed to rely on glutamine for proliferation, which yet remained to be further confirmed due to the very limited number of cell lines available for this study (Supplementary Fig. 1l). In line with this result, selective inhibition of glucose metabolism using UK5099, a potent inhibitor of the mitochondrial pyruvate carrier, preferentially inhibited the growth of

**Fig. 1** Oncogenic RTK differentially reprogram metabolic phenotypes. **a** Immunoblotting analysis. Cells were treated with indicated RTK inhibitors (100 nM) for 1 h. **b** IL3 dependence analysis. Cell growth fold changes with or without IL3 were plotted by counting cell numbers. Data were means of triplicates; error bars represented SD. **c** Cell sensitivity to RTK inhibition. Cells were treated with indicated RTK inhibitors for 72 h and cell viability was analyzed using CCK8 assay. Data were means of duplicates; error bars represented SD. **d** Oxygen consumption rate (OCR) and extracellular acidification rate (ECAR) measurement using Seahorse XF96 analyzer. Data were means of triplicates; error bars represented SD. **e** Heatmap depicting the metabolite intensities in the metabolomics data. Rows indicate different metabolites, and columns indicate different cells ($n = 3$ per cell line). The log transformed metabolite intensities are Z scored/standardized. **f** Principal component analysis (PCA). Three-dimensional clustergram depicts the internal structure of metabolomics data set with respect to variance by MetaboAnalyst 4.0. **g** Tracer scheme illustrating the flux of [U-$^{13}C_6$]-glucose (orange), [U-$^{13}C_5$]-glutamine (green) or [U-$^{13}C_{16}$]-palmitate (gray). Cells were cultured in the presence of [U-$^{13}C_6$]-glucose (12 h), [U-$^{13}C_5$]-glutamine (24 h) or [U-$^{13}C_{16}$]-palmitate (24 h) prior to mass spectrometry analysis. **h** Heatmap depicting representative $^{13}$C-labeled fraction contribution of the metabolite isotopologues. Rows indicate different metabolites and columns indicate different cells ($n = 3$ per cell line). The log transformed metabolite intensities are Z scored/standardized. **i** Glucose/glutamine dependence analysis. Cells were cultured in RPMI-1640 with or without glucose (GLC)/glutamine (GLN) for 3 days. Cell growth fold changes were plotted by counting cell numbers. Data were means of triplicates; error bars represented SD. **j** Transcriptome analysis. KEGG pathway enrichment analysis of differentially transcribed clusters in a heatmap of transcriptome profiling by RNA-seq. The different clusters are color-coded in Supplementary Fig. 1p. Bars show the enrichment score of pathways and are presented according to $p$ value using Fisher's exact test ($p < 0.05$). See the complete list of KEGG pathways in Supplementary Dataset 6. Source data are provided as a Source Data file

EGFR- and FGFR-dependent cancer cells (Supplementary Fig. 1m). Glutaminase inhibitor CB839 exhibited a more notable impact on the growth of RET-aberrant cancer cells (Supplementary Fig. 1n). Moreover, consistent with the fatty acid β-oxidation tracing (Fig. 1h), inhibition of fatty acid oxidation using etomoxir (ETO), an inhibitor of carnitine palmitoyl-transferase 1 A (CPT1A), only slightly affected the cell growth of all four genotypes (Supplementary Fig. 1o).

We also used RNA-sequencing (RNA-seq) to probe the genetic basis for the distinct metabolic dependency in these cells, which revealed the differentially transcribed gene clusters in these BAF3-RTK cell lines (Supplementary Fig. 1p, Dataset 6). Further KEGG pathway enrichment analysis of these distinguished clusters revealed overrepresented glycolytic and serine synthetic pathways in FGFR1- and EGFR-activated cells, respectively (Fig. 1j, Supplementary Dataset 6). To probe the clinical relevance of the observed transcriptional change in BAF3 cells, we analyzed the metabolic genes in EGFR- and FGFR-activated tumors described in Supplementary Fig. 1b. KEGG pathway enrichment analysis of altered metabolic genes (1.5-fold cutoff; $p < 0.01$) highlighted several metabolic pathways in EGFR- and FGFR-activated tumors respectively, such as pyruvate metabolism in FGFR amplified tumors and glycine serine and threonine metabolism in EGFR mutant tumors (Supplementary Fig. 1q). These findings overlapped with gene sets identified in the BAF3 cells. These data together demonstrated the preferentially reprogrammed metabolic phenotypes driven by corresponding RTK activation in cancer cells, which is worthy of in-depth investigation.

**EGFR activation promotes the serine synthesis pathway.** The data above suggested that EGFR activation mainly branched glycolysis to the serine synthesis pathway (SSP) (Fig. 1h). The upregulated serine metabolism was further confirmed in 740 lung adenocarcinoma patients extracted from the TCGA database by comparing EGFR mutated cancer patients ($n = 25$) with the rest wildtype EGFR tumors ($n = 715$) (Fig. 2a), suggesting that SSP could be a clinically-relevant vulnerability for this subtype of cancer. We hence treated RTK-driven cells with CBR5884, a widely used PHGDH inhibitor to decrease the de novo serine synthesis in these cells[21]. BAF3-EGFR but not BAF3-FGFR1 cells were responsive to PHGDH inhibition (Fig. 2b). Similar results were observed using NCT503, another reported PHGDH inhibitor[22] (Supplementary Fig. 2a). This observation was further confirmed in a panel of EGFR-aberrant cancer cells. Cancer cells with FGFR1/2/3 gene amplification and wildtype lung cancer cells

(except A431) without detectable driving gene alterations were tested in parallel as controls (Supplementary Dataset 5). We observed that PHGDH inhibition using CBR5884 showed the increased growth inhibition rate in EGFR mutant cells compared with cells with FGFR alteration. Wildtype cells, which lacked driving gene alterations, showed variable response but were mostly nonresponsive (Fig. 2c, Supplementary Fig. 2b). We also confirmed this metabolic vulnerability in vivo. PC9 xenograft model (Fig. 2d) and a NSCLC patient-derived xenograft (PDX) model LU-01-0251 with EGFR L858R mutation (Fig. 2e, Supplementary Dataset 7) were treated with EGFR inhibitor Gefitinib or NCT503, the only PHGDH inhibitor reported to exhibit in vivo activity. NCT503 treatment resulted in the significant tumor growth inhibition, slightly less potent than EGFR inhibition (Fig. 2d, e), but with minimum toxicity (Supplementary Fig. 2c, d). These results confirmed the SSP dependency as a metabolic vulnerability of EGFR-driven cancer. Very recent studies also reported the involvement of SSP in the acquired resistance to EGFR inhibitors[23,24], confirming the essential role of SSP in EGFR mutant cancer from a different perspective.

We further asked how increased serine production contributes to the malignant growth of EGFR-dependent cancer cells. To address this question, [U-$^{13}C_6$]-labeled glucose was used as a tracer to measure the fractional contribution to $^{13}$C-labeled nucleotide isotopologues in BAF3-EGFR cells, using BAF3-FGFR1 cells as a control (Fig. 2f). In this assay, we detected the enhanced proportion of M6 to M9 nucleotide isotopologues in EGFR-driven cells, which indicated the glycine and formate carbons incorporated into purine nucleotides (Fig. 2f, g, Supplementary Fig. 2e). In addition to nucleotide synthesis, glucose-derived serine could be incorporated into GSH via the generation of glycine, contributing to the redox homeostasis. We detected that glucose-derived GSH, shown as M2 isotopologue in the UHPLC-qTOF-MS spectrum using [U-$^{13}C_6$]-glucose tracer, was also enhanced by EGFR activation in BAF3 cells (Supplementary Fig. 2f). The impairment of SSP by knocking down PHGDH or phosphoserine phosphatase (PSPH) preferentially increased the reactive oxygen species (ROS) generation in EGFR mutant PC9 cells compared with the FGFR1 amplified DMS114 cells (Supplementary Fig. 2g, h). These results suggested a model that EGFR promoted SSP to supply building blocks for DNA/RNA synthesis and reducing equivalents. In support of this model, the growth of EGFR-activated cells was relatively less dependent on exogenous supply of serine, compared with FGFR1-activated cells (Fig. 2h).

We next proceeded to understanding how EGFR activation preferentially activated SSP. By comparing the expression of SSP

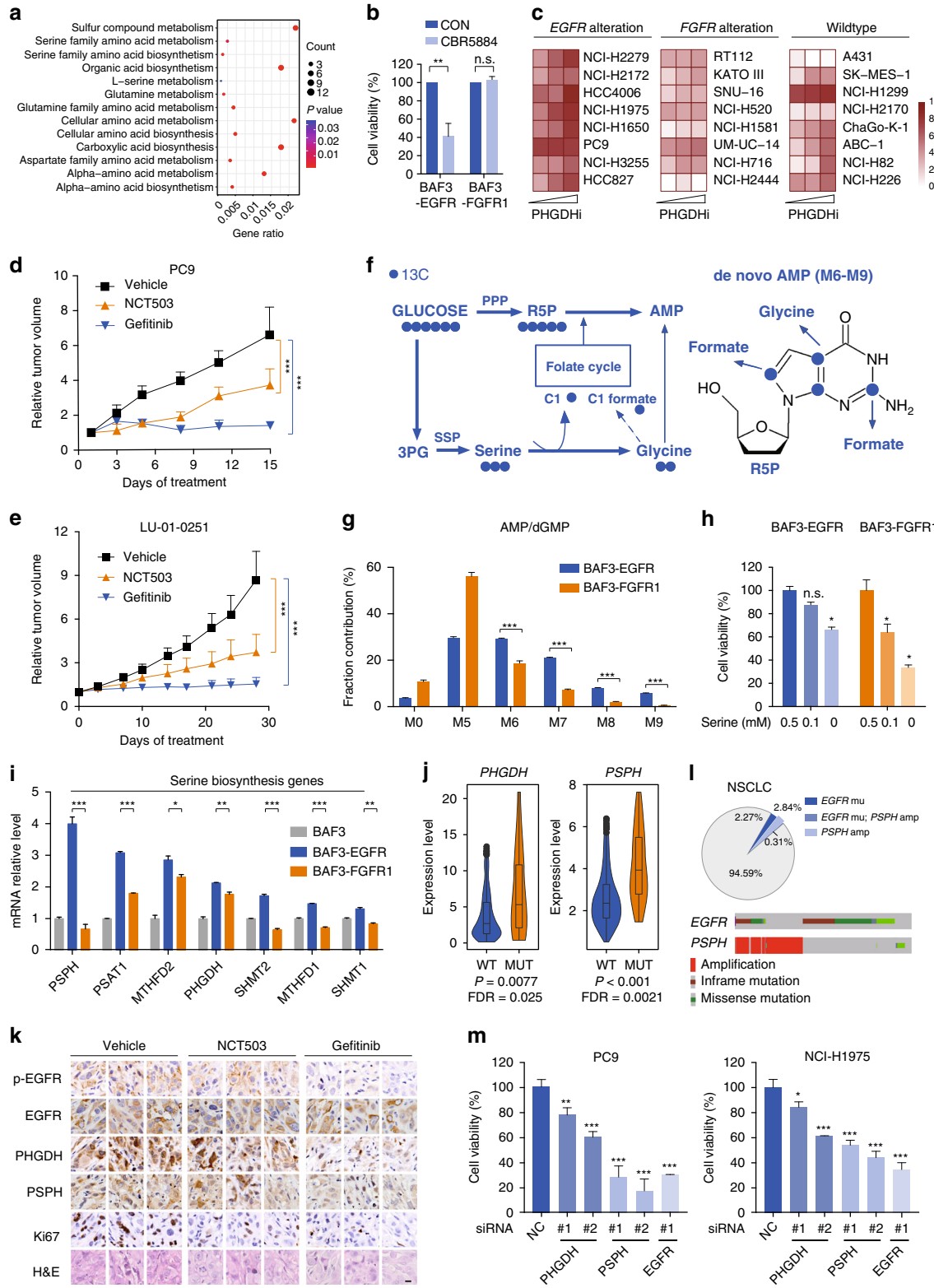

related enzymes between BAF3-EGFR and BAF3-FGFR1 cells, we discovered several key metabolic enzymes that were specifically upregulated upon EGFR activation, in particular PSPH, serine hydroxymethyltransferase 1 (SHMT1), SHMT2 and methylene-tetrahydrofolate dehydrogenase 1(MTHFD1) (Fig. 2i, Table 1). Among these enzymes, the significant upregulation of PHGDH and PSPH was also observed in the same set of lung adenocarcinoma patient samples, as described in Fig. 2a (Fig. 2j).

We also did the immunohistochemistry analysis of these metabolic enzymes in a panel of NSCLC PDX tumor tissues with mutant *EGFR* (n = 6) or wildtype RTK (n = 6). Compared with the wildtype tumors, most of the *EGFR* mutant tumors showed higher expression level of PSPH and PHGDH (Supplementary Fig. 2i). Importantly, the upregulated PHGDH and PSPH expression in the PDX model (LU-01-0251) could be reversed by the treatment of EGFR inhibitor (Fig. 2e, k).

**Fig. 2** EGFR activation promotes the serine synthesis pathway. **a** GO enrichment analysis of upregulated metabolic genes in *EGFR*-mutant lung cancer patients. Data were from 740 lung adenocarcinoma datasets from TCGA database (cutoff fold change > 2 and $p < 0.01$ versus *EGFR* wildtype cancer). Gene ratio represents the proportion of counted genes in the enriched pathway. **b** Sensitivity to PHGDH inhibition. Cell viability was measured using CCK-8 assay after treatment with CBR5884 (20 μM) for 72 h. Data were means of triplicates; error bars represented SD. **c** Sensitivity of a panel of cancer cells to PHGDH inhibition. Cells with indicated genetic alterations were treated with CBR5884 at 6.25, 12.5 or 25 μM for 6 days. Heatmap depicts the inhibition rate of the cell growth. **d**, **e** Tumor growth curve of PC9 xenograft and LU-01-0251 PDX. Mice were dosed with NCT-503 (40 mg/kg) or Gefitinib (5 mg/kg for PC9, 1 mg/kg for LU-01-0251) daily for indicated days ($n = 8$ for PC9, $n = 6$ for LU-01-0251). Data were means and error bars represented SEM. **f** Tracer scheme illustrating the flux of [U-$^{13}$C$_6$]-glucose to purine nucleosides. **g** $^{13}$C enrichment of purine nucleosides after [U-$^{13}$C$_6$]-glucose labeling for 24 h. **h** Serine dependence analysis. Cell viability was measured by counting cell numbers after 3-day culture. **i** Transcript analysis in BAF3-RTK cells compared with BAF3 cells. **j**, PSPH or PHGDH expression between *EGFR* mutant and wildtype subtypes. Data source was as described in **a**. **k** Immunohistochemistry analysis of tumor tissues from LU-01-0251 PDX as described in **e**. Tumor samples were collected at 6 h after the last dosing. Scale bar, 20 μm. **l** Upper: The occurrence of *EGFR* activating mutation and/or *PSPH* amplification in TCGA data sets; Lower: The alteration of *EGFR* and *PSPH* gene in NSCLC patients. **m** Cell growth measurement. Cells numbers were counted after transfected with indicated siRNAs for 72 h. Data were means of triplicates; error bars represented SD. For all bar graphs, $^{***}p < 0.001$, $^{**}p < 0.01$, $^{*}p < 0.05$, n.s. $\geq 0.05$ for two-tailed Student's *t* test. Source data are provided as a Source Data file

### Table 1 The sequences of primers used for RT-qPCR analysis

| Gene | Organism | Forward primer | Reverse primer |
|---|---|---|---|
| GLUT1 | Mus musculus | 5′-CTTCACTGTGGTGTCGCTGT-3′ | 5′-UUCUCCGAACGUGUCACGUTT-3′ |
| HK2 | Mus musculus | 5′-TGATCGCCTGCTTATTCACGG-3′ | 5′-AACCGCCTAGAAATCTCCAGA-3′ |
| PFKL | Mus musculus | 5′-GGAGGCGAGAACATCAAGCC-3′ | 5′-CGGCCTTCCCTCGTAGTGA-3′ |
| PKM2 | Mus musculus | 5′-GCCGCCTGGACATTGACTC-3′ | 5′-CCATGAGAGAAATTCAGCCGAG-3′ |
| LDHA | Mus musculus | 5′-TGTCTCCAGCAAAGACTACTGT-3′ | 5′-GACTGTACTTGACAATGTTGGGA-3′ |
| PHGDH | Mus musculus | 5′-CGGCAGAATTGGAAGAGAGGT-3′ | 5′-AGGAGTGGGGTATGGACAGTT-3′ |
| PSPH | Mus musculus | 5′-CATCTCTGGTGGCTTTCGGA-3′ | 5′-TTTCCTTTCCCACCCGACTC-3′ |
| PSAT1 | Mus musculus | 5′-GGTGTGATTTTCGCTGGTGC-3′ | 5′-AGGACTGATGGGCACTCTCT-3′ |
| SHMT1 | Mus musculus | 5′-CAGGGCTCTGTCTGATGCAC-3′ | 5′-CGTAACGCGCTCTTGTCAC-3′ |
| SHMT2 | Mus musculus | 5′-ATGCCCTATAAGCTCAATCCCC-3′ | 5′-TCTCATGCGTGCATAGTCAATG-3′ |
| MTHFD1 | Mus musculus | 5′-GCGGAGAGGATGAGATCATAGA-3′ | 5′-GTCACCCCGTCCACATCTT-3′ |
| MTHFD2 | Mus musculus | 5′-AGTGCGAAATGAAGCCGTTG-3′ | 5′-GACTGGCGGGATTGTCACC-3′ |
| GAPDH | Mus musculus | 5′-AGGTCGGTGTGAACGGATTTG-3′ | 5′-TGTAGACCATGTAGTTGAGGTCA-3′ |
| GLUT1 | Homo sapiens | 5′-TCACTGTCGTGTCGCTGTTT-3′ | 5′-GGCCACGATGCTCAGATAGG-3′ |
| LDHA | Homo sapiens | 5′-GGCCTGTGCCATCAGTATCT-3′ | 5′-GAAAAGGCTGCCATGTTGGA-3′ |
| PHGDH | Homo sapiens | 5′-CACGACAGGCTTGCTGAATGA-3′ | 5′-CTTCCGTAAACACGTCCAGTG-3′ |
| PSPH | Homo sapiens | 5′-GCATAAGGGAGCTGGTAAGTCG-3′ | 5′-ACCTGCATATTCACCGTTAAAGT-3′ |
| HIF1A | Homo sapiens | 5′-ACCTTCATCGGAAACTCCAAAG-3′ | 5′-CTGTTAGGCTGGGAAAAGTTAGG-3′ |
| MYC | Homo sapiens | 5′-GGCTCCTGGCAAAAGGTCA-3′ | 5′-CTGCGTAGTTGTGCTGATGT-3′ |
| ATF4 | Homo sapiens | 5′-ATGACCGAAATGAGCTTCCTG-3′ | 5′-GCTGGAGAACCCATGAGGT-3′ |
| NRF2 | Homo sapiens | 5′-TTCCCGGTCACATCGAGAG-3′ | 5′v-TCCTGTTGCATACCGTCTAAATC-3′ |
| β-actin | Homo sapiens | 5′-GGGACCTGACTGACTACCTC-3′ | 5′-ATCTTCATTGTGCTGGGTG-3′ |

Among these enzymes, PSPH, a key enzyme for the de novo serine synthesis as well as the rate-limiting enzyme of this pathway in the liver[25], was found to be significantly upregulated in *EGFR* mutant lung patients versus those bearing wildtype EGFR (Fig. 2j). Then, we were intrigued to look into the *PSPH* status in patients (1144 from TCGA database). This led to the unexpected identification of the co-occurrence of *PSPH* focal amplification and *EGFR* activating mutation (L858R point mutation, exon 19 deletion, or exon 21 deletion), which accounted for 16% (58/164, $p = 0.0067$) in *EGFR* mutant subtype (Fig. 2l). Considering the adjacent location of these two genes in Chr 7p 11.2, the concurrent occurrence of *PSPH* amplification and *EGFR* mutation may result from their physical linkage in this local region. The results suggested the existence of a set of NSCLC patients with highly promoted SSP due to the genetic alteration in both genes. These data may suggest PSPH as a target associated with EGFR activation in NSCLC. In line with this notion, knockdown of PSPH resulted in remarkably decreased cell growth of *EGFR*-mutant PC9 and NCI-H1975 cells, similar to EGFR inactivation (Fig. 2m, Table 2). In parallel testing of *FGFR* amplified cells showed that cell growth was differentially affected

(Supplementary Fig. 2j). Together, these data highlighted the SSP as a metabolic vulnerability in *EGFR*-mutant cancer (Supplementary Fig. 2k).

**FGFR activation enhances aerobic glycolysis and recycles lactate.** FGFR aberrations occur in multiple types of human cancers, including NSCLC, bladder cancer, and breast cancer etc.[26–28]. FGFR-activated cells appeared to avidly consume more glucose into glycolytic pathway (Fig. 1h). This was confirmed by the highly accelerated intracellular generation and extracellular secretion of lactate, the key product of aerobic glycolysis (Fig. 3a). Given the remarkable OCR enhancement (Fig. 1d), we suspected that lactate might serve as an alternative carbon source to fuel the TCA cycle. In support of this hypothesis, we detected the promoted fractional contributions of glucose to the TCA cycle intermediates in FGFR-activated cells by tracing the carbon enrichment of [U-$^{13}$C$_6$]-glucose to the intermediate metabolites of the TCA cycle in BAF3-RTK cells (Fig. 3b, Supplementary Dataset 4). The competitive uptake analysis of [U-$^{13}$C$_3$]-lactate (5 mM) in the presence of label-free glucose (10 mM) revealed that FGFR cells seemed to prefer to consume lactate for the TCA

**Table 2 siRNA targeting sequence**

| Gene | Organism | Sequence sense (5′−3′) |
|---|---|---|
| NC | Homo sapiens | UUCUCCGAACGUGUCACGUUTT |
| siEGFR | Homo sapiens | CUCCAGAGGAUGUUCAACAUTT |
| siFGFR1 | Homo sapiens | GACUUCACUGGUGUCAGAUTT |
| siPHGDH #1 | Homo sapiens | UAGCAAAGAGGAGCUGAUAUTT |
| siPHGDH #2 | Homo sapiens | GACUUCACUGGUGUCAGAUTT |
| siPSPH #1 | Homo sapiens | GGCAACAAGUCAAGGAUAAUTT |
| siPSPH #2 | Homo sapiens | GGAGUAUUGUAGUGAGCAUGUTT |
| siMYC #1 | Homo sapiens | CUCAACGUUAGCUUCACCAUTT |
| siMYC #2 | Homo sapiens | GUGCAGCCGUAUUUCUACUTT |
| siHIF1A #1 | Homo sapiens | CUCCCUAUAUCCCAAUGGAUTT |
| siHIF1A #2 | Homo sapiens | CGAGGAAGAACUAUGAACAUTT |
| siATF4 #1 | Homo sapiens | CUCCCAGAAAGUUUAACAAUTT |
| siATF4 #2 | Homo sapiens | CUGCUUACGUUGCCAUGAUT |
| CBFB #1 | Homo sapiens | GAAGCAAGUUCGAGAACGAUTT |
| CBFB #2 | Homo sapiens | CAGGAACCAAUCUGUCUCUUTT |
| CBFB #3 | Homo sapiens | CAGGCAAGGUAUAUUUGAAUTT |
| CEBPA #1 | Homo sapiens | CCUUCAACGACGAGUUCCUTT |
| CEBPA #2 | Homo sapiens | CGGUGGACAAGAACAGCAAUTT |
| CEBPA #3 | Homo sapiens | GCUGACCAGUGACAAUGACUTT |
| CTCF #1 | Homo sapiens | GGUGGAGCACUAGAACAAUTT |
| CTCF #3 | Homo sapiens | GUGCAAUUGAGAACAUUAUTT |
| CTCF #2 | Homo sapiens | GGUCUGCUAUCAGAGGGUUTT |
| E2F4 #1 | Homo sapiens | CGGCGGAUUUACGACAAUUTT |
| E2F4 #2 | Homo sapiens | CACCGGAGAUUUGCUCCAUTT |
| E2F4 #3 | Homo sapiens | CGGGAGACCACGAUUAUAUTT |
| ETS1 #1 | Homo sapiens | GUGGUUUCCAGUCCAAUUAUTT |
| ETS1 #2 | Homo sapiens | GUCCCACUAUUAACUCCAAUTT |
| ETS1 #3 | Homo sapiens | CGCUAUACCUCGGGAUUACUTT |
| FOS #1 | Homo sapiens | GGGAUAGCCUCUCUUACUAUTT |
| FOS #2 | Homo sapiens | GACAGACCAACUAGAAGAUTT |
| FOS #3 | Homo sapiens | CAAGGUGGAACAGUAAUGUTT |
| FOXO1 #1 | Homo sapiens | CCUACACAGCAAGUUCAUUTT |
| FOXO1 #2 | Homo sapiens | CCAUGGACAACAACAGUAAUTT |
| FOXO1 #3 | Homo sapiens | GCUCAAAUGCUAGUACUAUTT |
| GATA4 #1 | Homo sapiens | GUAGAUAUGUUUGACGACUTT |
| GATA4 #2 | Homo sapiens | GCCUCUCACCAAGAUGAAUTT |
| GATA4 #3 | Homo sapiens | GAAUAAAUCUAAGACACCAUTT |
| GTF2B #1 | Homo sapiens | CCAAGAGUCACAUGUCCAAUTT |
| GTF2B #2 | Homo sapiens | GGUUGUAGGUGACCGGGUUTT |
| GTF2B #3 | Homo sapiens | GCAGUUCGUAGUGAUGGAAUTT |
| HNF4A #1 | Homo sapiens | ACACCACCCUGGAAUUUGAUTT |
| HNF4A #2 | Homo sapiens | CAUGUACUCCUGCAGAUUUTT |
| HNF4A #3 | Homo sapiens | GCAGCUGCUGGUUCUCGUUTT |
| IRF5 #1 | Homo sapiens | GACGGAGAAACACCAACAUTT |
| IRF5 #2 | Homo sapiens | CGAGGAAGAAGCUCAUUAUTT |
| IRF5 #3 | Homo sapiens | GCAUGGUGGAGCAAUUCAAUTT |
| MAF #1 | Homo sapiens | GAACUGGCAAUGAGCAACUTT |
| MAF #2 | Homo sapiens | CUGGAAGACUACUACUGGAUTT |
| MAF #3 | Homo sapiens | GACGCGUACAAGGAGAAAUTT |
| MEF2A #1 | Homo sapiens | GGAGGACAGAUUCAGCAAAUTT |
| MEF2A #2 | Homo sapiens | GGGAAUGGAUUUGUAAACUUTT |
| MEF2A #3 | Homo sapiens | GCCCUUCUGUAAAGCGAAUUTT |
| MITF #1 | Homo sapiens | CCACCAAGUACCACAUACAUTT |
| MITF #2 | Homo sapiens | GUGGACUAUAUCCGAAAGUUTT |
| MITF #3 | Homo sapiens | GACCUAACCUGUACAACAAUTT |
| NOTCH1 #1 | Homo sapiens | GUCCAGGAAACACAUGCAAUTT |
| NOTCH1 #2 | Homo sapiens | GGGAGCAUGUGUAACAUCAUTT |
| NOTCH1 #3 | Homo sapiens | GGGCUAACAAAGAUAUGCAUTT |
| NR1H2 #1 | Homo sapiens | CCCAGAUCCCGAAGAGGAAUTT |
| NR1H2 #2 | Homo sapiens | CCAGCUAACAGCGGCUCAAUTT |
| NR1H2 #3 | Homo sapiens | GCCUGCAGGUGGAGUUCAUUTT |
| NFIC #1 | Homo sapiens | CCGACUUCCAGGAGAGCUUTT |
| NFIC #2 | Homo sapiens | CCACGAGUAGCAGCCGCAAUTT |
| NFIC #3 | Homo sapiens | GCAACUGGACGGAGGACAUTT |
| PPARG #1 | Homo sapiens | ACUCCACAUUACGAAGUACUTT |
| PPARG #2 | Homo sapiens | CCUCAUGGCAAUUGAAUGUTT |
| PPARG #3 | Homo sapiens | CUGGCCUCCUUGAUGAAUAUTT |
| SOX2 #1 | Homo sapiens | CUGCAGUACAACUCCAUGAUTT |
| SOX2 #2 | Homo sapiens | CCACCUACAGCAUGUCCUAUTT |
| SOX2 #3 | Homo sapiens | GGACAUGAUCAGCAUGUAUTT |
| SREBF1 #1 | Homo sapiens | GGAGGCUUCUCUACAGGAAUTT |
| SREBF1 #2 | Homo sapiens | CCUUGGUGCUUCUCUUUGUTT |
| SREBF1 #3 | Homo sapiens | GCCUGACCAUCUGUGAGAAUTT |
| TCF7 #1 | Homo sapiens | GCAUGUACAAAGAGACCGUTT |
| TCF7 #2 | Homo sapiens | CCACCCAUCCUUGAUGCUAUTT |
| TCF7 #3 | Homo sapiens | CCGCAACCUGAAGACACAAUTT |
| TEAD1 #1 | Homo sapiens | CUGCCAUUCAACAAGAGAUUTT |
| TEAD1 #2 | Homo sapiens | GGCAUGCCAACCAUUUCUUAUTT |
| TEAD1 #3 | Homo sapiens | GUGGUAACAAACAGGGGAUUTT |

supply in vitro (Fig. 3c, Supplementary Fig. 3a). We further traced the fractional contributions of glucose and lactate to the TCA cycle in *FGFR1* amplified NCI-H1581 xenograft model, in which mice were co-injected with [U-$^{13}C_6$]-glucose and [3-$^{13}C$]-lactate intravenously and tumor tissues were collected after 30 min. Glucose carbons incorporated into the intermediates of TCA cycle were similar to lactate carbons (Fig. 3d), after normalized by the plasma lactate M3 or M1 isotopologue intensity resulted from peripheral conversion (Supplementary Fig. 3b). These results highlighted lactate as an equal fuel for the TCA cycle as glucose. In line with these results, inhibition of lactate production using Oxamate or GSK2837808A[29] decreased OCR level in *FGFR1* amplified NCI-H1581 cells, similar to the impact of FGFR inhibition (Fig. 3e). In contrast, the intervention of lactate production only slightly affected mitochondrial capacity in EGFR-dependent PC9 cells (Supplementary Fig. 3c).

Accelerated oxidative phosphorylation is known to mainly supply ATP for the rapid malignant growth, resulting in large amounts of ROS as a byproduct. We further measured the contribution of lactate to ATP and ROS production in FGFR-activated cells. In BAF3 cells, activation of FGFR1 was associated with higher level of ATP and ROS generation compared with EGFR activation (Fig. 3f, g). Both ATP and ROS production in FGFR1-activated NCI-H1581 cells could be partially reversed by LDH inhibitors, with an extent similar to FGFR inhibition (Fig. 3f, g). These data together revealed that lactate plays an important role in fueling oxidative phosphorylation in FGFR-aberrant cancer.

We also compared the expression of glycolytic enzymes upon FGFR1 or EGFR activation in BAF3 cells. A few glycolytic enzymes, such as lactate dehydrogenase A (LDHA), ATP-dependent 6-phosphofructokinase (PFKL), glucose transporter 1 (GLUT1) and hexokinase 2 (HK2) were found particularly upregulated in BAF3-FGFR1 cells (Fig. 3h). *FGFR* amplification associated upregulation of these enzymes were confirmed in 740 lung adenocarcinoma patient samples extracted from TCGA database (Fig. 3i). Using immunohistochemistry analysis, these findings were recapitulated in NSCLC PDX tumor tissues with *FGFR1/2* amplification versus RTK wildtype tumors. Most tumors with *FGFR* alteration showed higher expression of LDHA and HK2 (Fig. 3j, Supplementary Fig. 3d). Furthermore, we used Project Achilles, a CRISPR/Cas9 screening-based dataset[30,31], to reveal the cell growth dependence on these metabolic genes. These results indicated that the dependence on LDHA, PFKL and PKM was significantly higher in *FGFR*-amplified cancer cells than the *FGFR* wildtype cancer cells. As a control, the dependence on PHGDH and PSPH showed no difference between the indicated subgroups (Supplementary Fig. 3e), suggesting that *FGFR*-amplified cells were highly dependent on glycolysis.

All these findings suggested the essential role of lactate production in FGFR-driven cells. We hence disrupted the lactate production in a panel of cancer cells, as described in Fig. 2c using GSK2837808A, a highly selective and potent LDHA inhibitor. The growth of FGFR-aberrant cells appeared more responsive to LDH inhibitor compared with the control cells (Fig. 3k), suggesting that lactate metabolism was preferentially required for FGFR-driven malignant growth. Importantly, lactate production inhibitor Oxamate significantly inhibited the tumor growth without mice body weight change in SNU16 and NCI-H1581 xenograft models (Fig. 3l, Supplementary Fig. 3f), along with the decreased lactate and citrate generation in the tumors detected by [U-$^{13}C_6$]-glucose tracer (Fig. 3l). We further tested the efficacy of Oxamate in two *FGFR2*-amplified NSCLC PDX models. In model LU6429, the benefit of either Oxamate or AZD4547 was clear but with different extents between the individual tumors. Of note, combinational inhibition of LDH and FGFR using AZD4547-

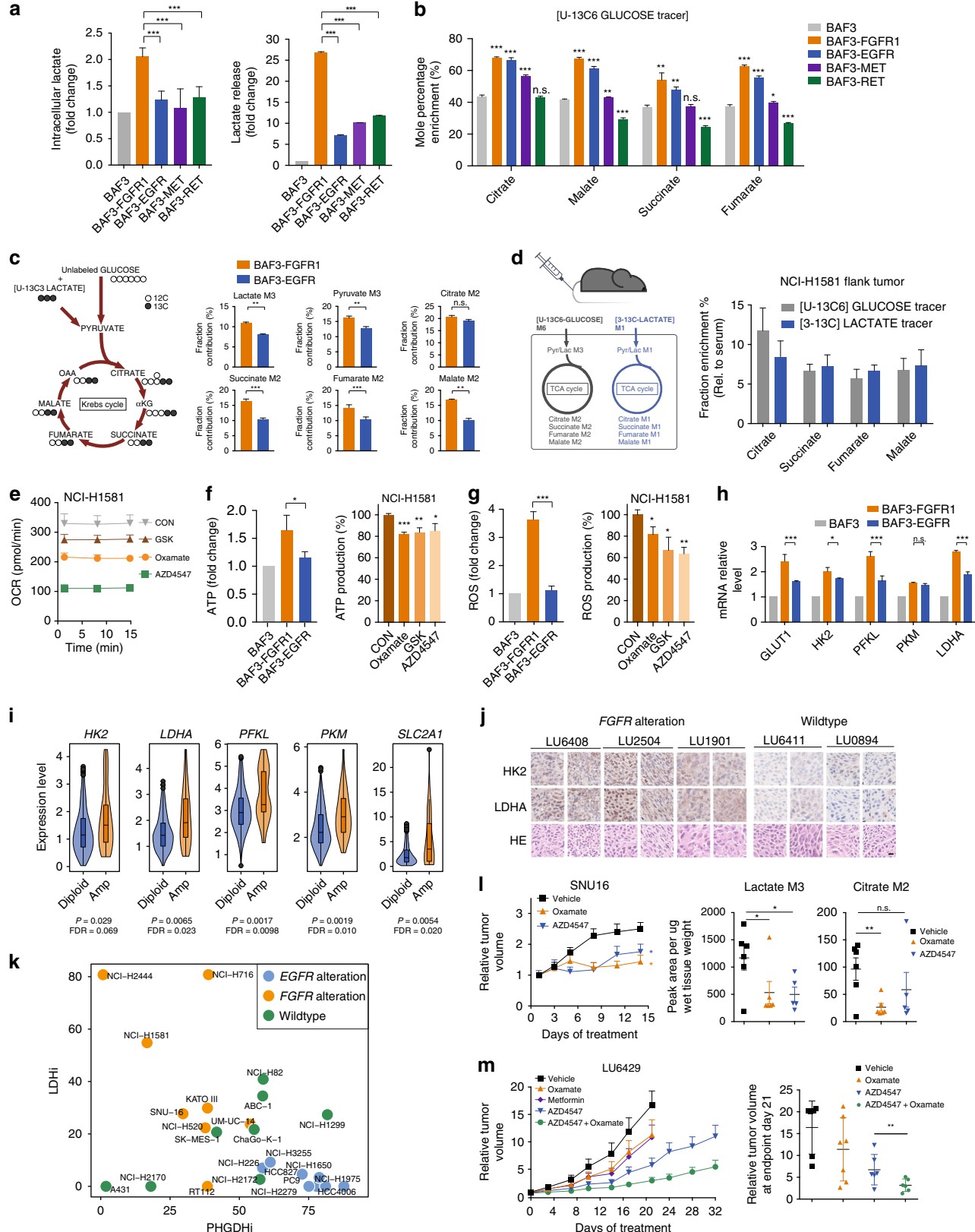

Oxamate combination caused a more universal and sustainable benefit in tumor growth inhibition than FGFR inhibition alone (Fig. 3m, Supplementary Fig. 3g), suggesting the potential in lessening the resistance development to FGFR inhibitors. Of note, the combination study was well tolerated as indicated by mice body weight change. Similar trend was observed in another

*FGFR2*-amplified model LU0743 though the therapeutic effect appeared less striking (Supplementary Fig. 3h). Of interest, we also used metformin to block the mitochondrial respiration, given the recycled lactate was utilized to fuel the oxidative phosphorylation. Indeed, metformin showed similar efficacy compared with the inhibition of lactate production (Fig. 3m).

**Fig. 3** FGFR activation enhances aerobic glycolysis and recycles lactate. **a** Lactate production and lactate release. **b** Enrichment of $^{13}$C-labled intermediate metabolites. Cells were cultured in the presence of [U-$^{13}$C$_6$]-glucose for 24 h. **c** Competitive uptake of glucose and lactate. Cells were cultured with both [U-$^{13}$C$_3$]-lactate (5 mM) and label-free glucose (10 mM) for 24 h. Left, Tracing map of [U-$^{13}$C$_3$]-lactate. Right, Enrichment of $^{13}$C-labled intermediate metabolites. **d** The incorporation percentage of lactate to the TCA cycle in NCI-H1581 xenograft tumors ($n = 6$). Left, Scheme for co-infusions of [U-$^{13}$C$_6$]-glucose and [3-$^{13}$C]-lactate and the tracing map. Right, Fraction enrichment of the intermediates. **e** OCR measurement. Cells were treated with AZD4547 (100 nM, 24 h), Oxamate (10 mM, 6 h) or GSK2837808A (20 μM, 6 h). **f, g** ATP production and ROS level. NCI-H1581 cells were treated as indicated in **e**. **h** Transcript analysis of BAF3-RTK cells normalized by that in BAF3 cells. **i** The comparison of glycolytic gene expression between *FGFR* amplified and diploid cancer. Data were from 740 lung adenocarcinoma patients in TCGA data sets. **j** Immunohistochemistry analysis of tumor tissues from NSCLC PDX tumors. Shown are the representative fields from one section per tumor tissue (2 independent tumor tissues per PDX model). Scale bar, 20μm. **k** Scatter plot showing the inhibition rate of PHGDH inhibitor (PHGDHi) CBR5884 (12.5 μM) and LDH inhibitor (LDHi) GSK2837808A (50 μM). Cells were treated for 6 days. **l** Left: tumor growth curve of SNU16 xenograft model. Right: glucose-derived metabolites in tumor tissue. See the dosing regimen in Methods. Data were means ($n = 6$) and error bars represented SEM. **m** Tumor growth curve of LU6429 PDX model and grouped scatter plot of individual mice relative tumor volume on Day 21. See the dosing regimen in Methods. Data in **a**, **b**, **e**-**h** were means of triplicates and error bars represented SD. Data in **c**, **d**, **l**, **m** were means and error bars represented SEM. For all bar graphs, $^{***}p < 0.001$, $^{**}p < 0.01$, $^{*}p < 0.05$, n.s. $\geq 0.05$ for two-tailed Student's *t* test. Source data are provided as a Source Data file

Together, we herein demonstrated the importance of lactate production and consequential oxidative phosphorylation in FGFR-aberrant cancer and indicated therapeutic promise arising from this metabolic phenotype (Supplementary Fig. 3i).

Thus far, we have shown that *EGFR* and *FGFR* gene alteration could be used to stratify tumors responsive to inhibitors of SSP and lactate production, respectively. To strengthen this result, we also tested several tumor models bearing wildtype RTK. In A431 cell xenograft, which often has been used as a control for RTK study[32], PHGDH or LDH inhibitors were barely responded (Supplementary Fig. 3j). Likewise, in 3 NSCLC PDX models without detectable driving gene alterations (LU-01-0393, LU2071 and LU-01-0416), we did not observe the apparent therapeutic benefits (Supplementary Fig. 3k). The results together emphasized the importance of patient selection for these inhibitors.

**ATF4 and c-Myc orchestrate metabolic reprogramming.** The remaining question is what accounts for the preferential metabolic rewiring in EGFR and FGFR cells. Previous evidence has highlighted the role of transcriptional factors (TFs) as key nodes in rewiring the metabolic network in cancer cells[33–36]. We hence proceeded to identify TFs responsible for orchestrating the transcriptome reprogramming in FGFR- and EGFR-dependent cells. According to the RNA-seq data in BAF3 cells, we stratified metabolic genes (annotated by KEGG database) whose expression were promoted by RTK activation (1.5-fold, $p < 0.01$ cutoff in relative to parental cells), and established a network model describing the TF-target interactions according to the published TF database (Cistrom, ORegAnno, mSigDB, CellNet and UCSC). This led to the identification of 127 and 138 TFs that might be potentially involved in regulating the differentially transcribed metabolic enzymes in EGFR- and FGFR-addicted context respectively (Supplementary Dataset 8). Based on this bioinformatic annotation, we carried out a functional screen. Among the TFs suggested by the TF-target interaction network, we knocked down 25 representing TFs in FGFR3-dependent RT112 and EGFR-activated PC9 cells, and the expression of representative metabolic genes, *PSPH* and *PHGDH* for the SSP and *LDHA* and *GLUT1* for glycolysis, were examined by RT-qPCR analysis respectively. This identified ATF4, FOXO1, HIF1A, MAF, MYC, and SREBP1 in FGFR-dependent cancer cells whereas ATF4 and MYC in EGFR-addicted cancer cells, which were required for the transcription of the signature genes in glycolysis or SSP (Fig. 4a, Supplementary Dataset 9). Among these TFs, we noticed that only HIF1A and MYC expression were affected by FGFR inhibition. Likewise, EGFR inhibitor Gefitinib could affect both ATF4 and MYC expression in *EGFR* mutant cancer cells (Fig. 4b) and xenograft models (Fig. 4c). Further, knockdown of MYC rather

than ATF4 could substantially suppress the cell growth of FGFR3-activated RT112 cells, whereas in EGFR-aberrant PC9 cells, only knockdown of ATF4 clearly suppressed cell growth (Fig. 4d). All these data concluded the essential role of HIF1A-MYC (in FGFR-addicted cells) and ATF4-MYC (in EGFR-activated cells) in transcriptionally regulating the metabolic network, and MYC and ATF4 seemed to play a more dominant role for FGFR-addicted cells and EGFR-activated cells, respectively.

We then looked into the network analysis to confirm the close association of the identified TF pairs in regulating metabolic network. Ten altered metabolic genes were co-regulated by MYC and ATF4 in EGFR cells, including PSPH, while over 20 genes encoding metabolic enzymes are co-regulated by HIF1A and MYC in FGFR cells including LDHA (Fig. 4e). To carefully dissect the role of these TF pairs in both cell contexts, we noticed that MYC depletion could downregulate HIF1A expression whereas MYC was intact upon HIF1A depletion, further supporting the dominant role of MYC in FGFR cells (Fig. 4f, Supplementary Fig. 4a). Similarly, in EGFR-constitutively activated PC9 cells, ATF4 appeared playing a more dominant role as knockdown of ATF4 resulted in the downregulation of MYC, not the case vice versa (Fig. 4f, Supplementary Fig. 4a). We then confirmed the impact of these TFs on the metabolic phenotypes. Consistent with results shown above, intracellular production of pyruvate and lactate in FGFR-activated cancer cells were decreased by MYC rather than ATF4 depletion (Fig. 4g). In parallel, in a $^{13}$C$_6$-glucose-labeled ISA assay, knockdown of ATF4 instead of MYC significantly inhibited serine and nucleotide synthesis in PC9 cells, similar to EGFR inhibition (Fig. 4h, Supplementary Fig. 4b, c). Together, we herein established the RTK-initiated transcription regulatory network, which highlighted transcriptional factors ATF4 and MYC as the key nodes in preferentially reprogramming the metabolic network in cancers with aberrant EGFR or FGFR (Fig. 4i).

**Discussion**

Currently, apart from a few cases with autonomous genetic abnormalities in metabolic enzymes[12–14], the metabolic vulnerabilities of most cancers remain unclear. With the advancement of metabolic inhibitor discovery, it is imperative to understand patient stratification strategy for the treatment. Lately, a few studies have revealed the new therapeutic opportunities stemmed from the identification of metabolic vulnerabilities. For example, triple-negative breast cancer (TNBC) was highly dependent on de novo pyrimidine synthesis. Inhibition of pyrimidine synthesis could sensitize TNBC to chemotherapy[37]. Likewise, glutathione biosynthesis was discovered as a metabolic vulnerability in PI(3)

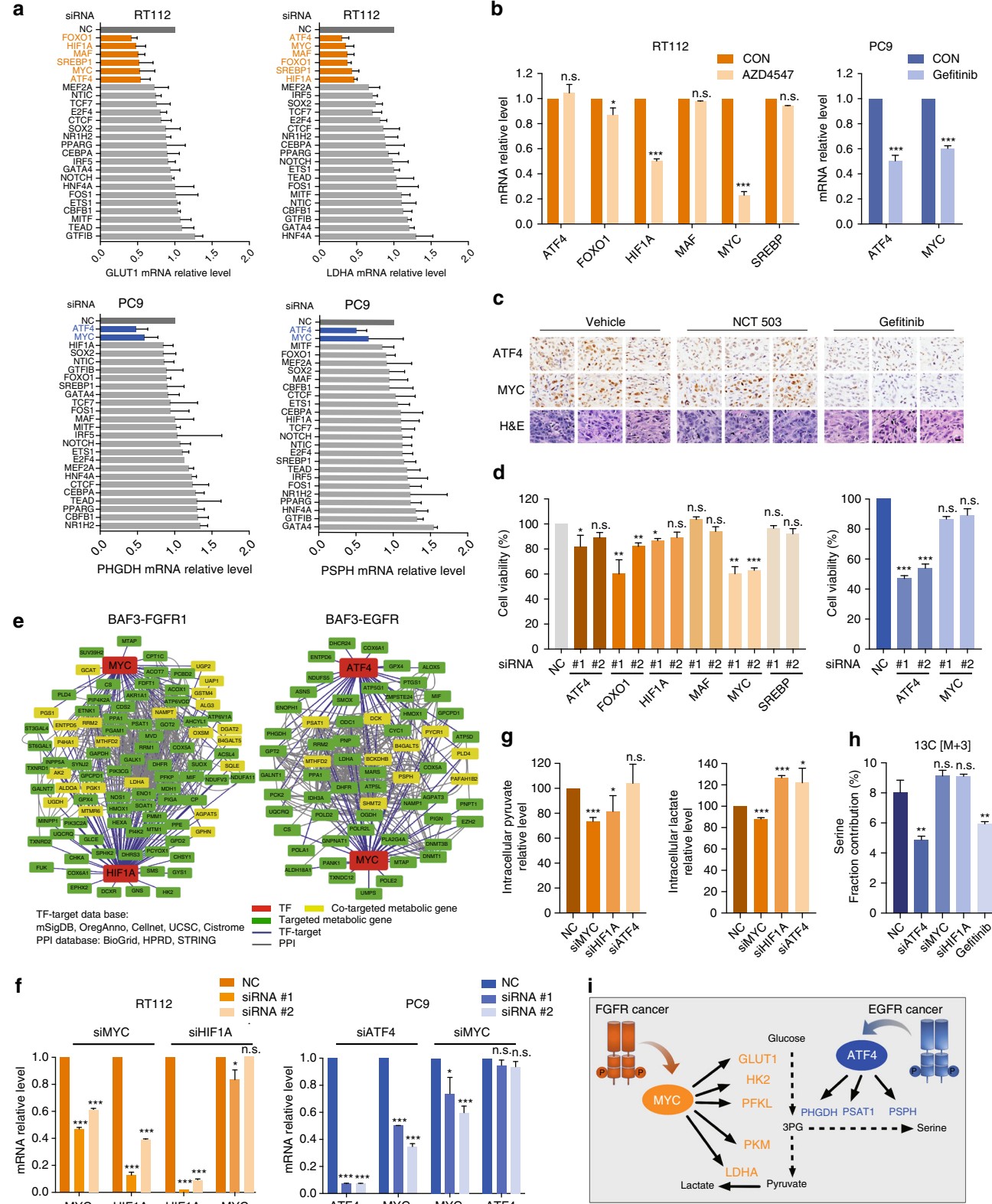

K/Akt-driven breast cancer[38] and purine synthesis for small cell lung cancer[39]. All these findings showed the possibility of precisely directing the metabolic therapeutics to responsive cancer subtypes defined by non-metabolic signatures.

Regardless of these advancements, the efficacious therapies towards metabolic vulnerabilities of human cancer remains very limited. Quite a few metabolic inhibitors are undergoing clinical trials, but the strategy for patient stratification is unclear. In this study, we chose to establish the linkage between metabolic vulnerabilities and the well-validated cancer genotypes. We took an integrative approach of metabolomics and transcriptomics to identify the metabolic vulnerabilities of RTK-driven cancer. These efforts allowed us to define the metabolic preference of RTK-aberrant cancer, showing that RTKs alterations could be used for

**Fig. 4** ATF4 and c-Myc orchestrate metabolic reprogramming. **a** Metabolic gene expression upon the depletion of candidate transcription factors (TFs). The $Y$-axis indicates the transcription factors and X-axis shows relative mRNA levels of metabolic genes. Gene expression was measured by RT-qPCR after transfected with indicated siRNA for 72 h. NC, negative control. Data were means of duplicates; error bars represented SD. **b** RT-qPCR transcript analysis. Cells were treated with AZD4547 (100 nM) or Gefitinib (100 nM) for 24 h and the mRNA level of indicated TFs was measured by RT-qPCR. The expression level of indicated genes was normalized by that of the untreated group (CON). Data were means of triplicates; error bars represented SD. **c** Immunohistochemistry analysis. PC9 xenograft models were treated with NCT-503 (40 mg/kg) or Gefitinib (5 mg/kg) daily for 15 days (n = 8). Tumor samples were collected at 6 h after last dosing. Scale bar, 20 μm. **d** Cell viability assay. Cells were transfected with two independent siRNAs targeting indicated TFs for 72 h and the cell viability was analyzed by counting cell numbers. Data were means of duplicates; error bars represented SD. **e** A network model describing the TF-targeted metabolic genes interactions. **f** RT-qPCR transcript analysis. Cells were transfected with indicated siRNA for 72 h and metabolic gene expression was measured by RT-qPCR. Data were means of triplicates; error bars represented SD. **g** Intracellular pyruvate and lactate levels. RT112 cells were transfected with indicated siRNA for 48 h and pyruvate and lactate aboundance was measured using GC/MS. **h** Analysis of [U-$^{13}C_6$]-glucose-derived serine. PC9 cells were transfected with indicated siRNA for 48 h followed by 24 h-culture in the presence of [U-$^{13}C_6$]-glucose. Serine isotopologue M3 was measured by GC-MS. Gefitinib treatment (100 nM, 24 h) was used as a positive control. Data in **g**, **h** were means of triplicates; error bars represented SEM. **i** Diagram depicting RTK-driven transcriptional reprogramming to orchestrate metabolic changes. For all bar graphs, ***$p <$ 0.001, **$p < 0.01$, *$p < 0.05$, n.s. ≥ 0.05 for two-tailed Student's $t$ test. Source data are provided as a Source Data file

patient stratification for metabolic inhibitors. For example, *EGFR* gene activating alterations, which have been included in routine diagnosis for NSCLC patients, could be a molecular signature for stratifying responsive tumors for SSP inhibitors. Likewise, LDH inhibitors, which have been extensively explored in pre-clinical and clinical studies, could be delivered to cancer patients with *FGFR* gene alterations. Of note, in-depth analysis of clinical samples extracted from TCGA database suggest that the metabolic phenotypes identified in this study may cross histological types bearing the same oncogenic RTK alteration, suggesting broad benefit that is worthy of clinical tests.

Oncogene-driven metabolic alterations have been observed in cancer cells, which is generally considered as cell-autonomous adaptations. According to our results, different oncogenic alterations are associated with heterogeneous metabolic vulnerabilities, which mechanistically stems from the differential transcriptome orchestrated by defined transcriptional factors. Our findings confirm that oncogene activation plays a causative, rather than correlative, role in reprogramming metabolic phenotypes in malignant cells. Oncogene activation often results in the rewiring of a complex metabolic network. For instance, gene abnormalities in KRAS, one of the most frequently mutated oncogenes in human cancers, alter cellular glucose uptake, glycolytic flux, and glutamine usage. The identification of the metabolic vulnerabilities in a complex network will require better understanding of intrinsic metabolic preference with distinct nutrients and pathways for rapid growth[40]. Most recently, it has been reported that glucose fuels the TCA cycle via lactate recycle and turnover in human NSCLC[41,42]. Together with the existing understanding, our data pinpoint the crucial role of glycolysis and resultant lactate production, which fuels TCA cycle for energy production in FGFR-aberrant cancer.

In this study, we have discovered that FGFR oncogenic signaling mainly converges on MYC for transcriptional regulation of downstream glycolytic enzymes whereas EGFR activation preferentially utilizes ATF4 to drive metabolic network reprogramming. Of note, a previous work from Cantley's group, which was intrigued by the frequent occurrence of transcription factor *NRF2* activation mutation in NSCLC, has demonstrated that upregulation of NRF2 induces the expression of the key serine/glycine biosynthesis enzymes via ATF4[43]. Consistently, a previous work discovered that in *EGFR* mutant cancer, knockdown of NRF2 remarkably decreased the ATF4 expression as well as downstream metabolic enzymes[44]. These results positioned NRF2 upstream of ATF4 in *EGFR* mutant context in NSCLC cells. Interestingly, *NRF2* mutation occurs often in NSCLC but appears very rare in *EGFR* mutant cancer. According to TCGA database, among 1940

lung carcinoma patients (including 44 *EGFR* mutant patients), 106 patients showed *NRF2* mutation yet none of these patients co-occurred with *EGFR* mutation. We speculate that EGFR activation, in addition to NRF2 activating mutation or Keap1 dysfunction, might represent a complimentary mechanism for upregulating the serine biosynthetic pathway via NRF2-ATF4 axis in lung cancer.

Certainly, in addition to the complexity imposed by the oncogenotype, a full accounting of tumor metabolism must consider extrinsic influences. Metabolic adaptation of cancer cells is implemented in the stressful and dynamic microenvironment, where concentrations of crucial nutrients such as glucose, glutamine and oxygen are spatially and temporally heterogeneous. It should be noted that while cancer cell lines are cultured in various culture medium supplied with different nutrients, cancer cell lines bearing same oncogenic driving mutation seemed to exhibit the similar metabolic vulnerability in cell-based assays. We reason that oncogenic driving mutation exhibit a dominate impact on the cell intracellular metabolic homeostasis. Consistently, the observed growth dependency in EGFR- and FGFR-driven cancer could be recapitulated in vivo including PDX models that are believed to faithfully reserve the clinical features of patients. We suspect that the potential translation may be only applicable to specific oncogene-addicted cancer, in which oncogene activation alone largely dictates the fate of tumor cells. However, other mutations, such as KRAS activation and TP53 deletion in the pancreas may result in the metabolic behaviors of cancer that appears to be highly context dependent compared to the lung organ[6]. Of great interest, in our cell models, MET-driven cancer cells barely show unique metabolic features in vitro, which was consistent with the analysis of clinical samples (Supplementary Fig. 1b). In a previous report, MET has been demonstrated to promote glutamine synthesis from glucose flux in liver tumors which were dictated by environmental factors in vivo[4]. This may suggest the importance of understanding the role of MET in tumor environment and echoed the increasing insights of its involvement in reprogramming tumor microenvironment.

In summary, our data together pinpoint the crucial role of glycolysis and resultant lactate production, which fuels the TCA cycle for energy production in FGFR aberrant cancer, and the serine synthesis for nucleotide biosynthesis and redox homeostasis in EGFR constitutively activated cancer. This study may advance the current understanding in the field in two aspects. Firstly, oncogenic RTK-driven metabolic reprogramming could result in distinct metabolic vulnerabilities that could be exploited for cancer treatment. Secondly, heterogeneous metabolic dependency arises from the differential expression of

metabolic genes orchestrated by transcription factors. These findings have the important translational value for guiding the patient stratification of metabolic inhibitors.

## Methods

**Cell culture**. The information of all the cells were summarized in Supplementary Dataset 5. All the cell lines were regularly authenticated by analyzing the DNA profile of 8 short tandem repeat (STR) loci plus amelogenin (Genesky Biotechnologies Inc., Shanghai, China) and were maintained in appropriate culture medium as suppliers suggested. BAF3 cells (DSMZ, ACC 300) were cultured in the RPMI-1640 medium containing 10 ng/mL of IL3 (PeproTech, 213-12-50). BAF3-RTK cells were constructed in our laboratory[45,46], and maintained in the RPMI-1640 medium without IL3. Unless otherwise stated, in all assays involving BAF3 cells, BAF3 parental cells were cultured with IL3 and BAF3-RTK cells were cultured without IL3.

**Animal experiments**. Tumor-bearing mice were randomized into groups and started dosing when average tumor volume reached 80–150 mm³. Dosing details were described as below. Tumor growth was monitored by the measurement of tumor size using caliper every three days using the formula (length × width²)/2. The individual relative tumor volume (RTV) was calculated as follows: $RTV = Vt/V_0$, where Vt is the volume on each day and $V_0$ is the volume at the beginning of the treatment.

Dosing regimens for cell line-based xenograft and PDX models were as follows. For PC9 model, NCT503 (40 mg/kg, p.o.) and Gefitinib (5 mg/kg, p.o.) were given daily for 15 days (8 mice per group). For SNU16 model, Oxamate (750 mg/kg, i.p.) and AZD4547 (5 mg/kg, p.o.) were given daily for 14 days (6 mice per group). For NCI-H1581 model, Oxamate (750 mg/kg, i.p.) and AZD4547 (2.5 mg/kg, p.o.) were given daily for 21 days (5 mice per group). For A431 model, Oxamate (750 mg/kg, i.p.) and NCT503 (40 mg/kg, p.o.) were given daily for 13 days (10 mice per group). For LU-01-0251 PDX model, NCT503 (40 mg/kg, p.o.) and Gefitinib (1 mg/kg, p.o.) were given daily for 24 days (6 mice per group). For LU6429 PDX model, Oxamate (750 mg/kg, i.p.), Metformin (250 mg/kg, i.p.) and AZD4547 (10 mg/kg, p.o.) were given daily (6 mice per group except for Oxamate-treated group of 7 mice). For LU0743 PDX model, Oxamate (750 mg/kg, i.p.), and AZD4547 (10 mg/kg, p.o.) were given daily (6 mice per group). For LU-01-0393 and LU-01-0416 PDX models, Oxamate (750 mg/kg, i.p.), and NCT503 (40 mg/kg, p.o.) were given daily for indicated days (6 mice per group). For LU0271 model, Oxamate (750 mg/kg, i.p.), and NCT503 (40 mg/kg, p.o.) were given daily for 20 days (5 mice per group). The dosing period was determined by the endpoint tumor volume, and tumor samples were collected at 6 h after the last dosing. For all studies involving combination treatment, drugs were given separately. Growth curve was plotted by measuring the relative tumor volume three times per week compared to vehicle group. All the data were means and error bars represented SEM.

The experiments of A431, SNU16 and NCI-H1581 xenograft models were approved and performed according to the Institute Animal Care and Use Committee (IACUC) at Shanghai Institute of Materia Medica. The experiments of PC9 xenograft, LU-01-0251, LU-01-0393 and LU-01-0416 PDX models were approved and performed according to the IACUC at WuXi AppTec. The experiments of LU2071, LU0743, and LU6429 PDX models were approved and performed according to the IACUC at CrownBio. During all the studies, the care and use of animals were conducted in accordance with the regulations of the Association for Assessment and Accreditation of Laboratory Animal Care (AAALAC).

**Stable isotope resolved metabolomics in vivo**. The glucose utilization and transformation in SNU16 xenograft model was determined by bolus injection of [U-¹³C₆]-glucose tracer in the tail vein. 25% (w/v) sterile filtered [U-¹³C₆]-glucose (by 0.22 μm microporous filter membrane) in 0.9% NaCl was pre-prepared. Intravenous injection of [U-¹³C₆]-glucose for 200 μL in the tail vein of mice (20 g) at 6 h after the last dosing. After 45 min, the mice were killed immediately, and the tumor tissues were collected and preserved at −80 °C. Incorporation of ¹³C into metabolites extracted from tissues was profiled by GC-MS.

The contribution of lactate in vivo was determined by injection with [U-¹³C₆]-Glucose and [3-¹³C]-lactate concurrently in subcutaneous NCI-H1581 tumor-bearing mice. Sterile filtered the mixture of 25% (w/v) [U-¹³C₆]-glucose and 6.25% (w/v) [3-¹³C]-lactate (n/n = 2:1) (by 0.22 μm microporous filter membrane) in 0.9% NaCl was pre-prepared. Intravenous injection for 100 μL in the tail vein of mice (20 g). Short injections were used to limit labeling complexity. After 30 min, the mice were killed immediately, and the tumor tissues were collected and preserved at −80 °C. Incorporation of ¹³C into metabolites extracted from tissue was profiled by GC-MS.

**The analysis of TCGA patients**. For lung adenocarcinoma, LUAD dataset was extracted from TCGA and used for further analysis. The upregulated genes in EGFR mutant tumors (L858R, exon 19 deletion or exon 21 deletion) were determined compared to the counterpart with wildtype EGFR (cutoff fold-change > 1.5 and p < 0.05), and gene functions were enriched by R package ClusterProfiler using

annotation of GO-Biological Process. Gene amplifications were annotated by the cBioPortal public dataset through GISTIC algorithm, a widely used method able to differentiate the focal alteration and chromosome level copy number gain.

**Identification of key transcription factors regulating metabolic DEGs**. Differentially expressed genes (DEGs) were identified by comparing gene expression in parental BAF3 and BAF3-RTK cells. DEGs were selected with a cutoff of 1.5 fold-change and p < 0.01 using two-tailed Student's t test. Among DEGs, metabolic genes were identified by KEGG database.

To identify key TFs regulating metabolic DEGs, 1118148 TF-targets interaction data pointing for 2160 TFs were collected from public databases including Cistrom, mSigDB, OregAnno, CellNet and UCSC. The TFs were enriched using Fisher's exact test and p-values were adjusted using Benjamini-Hochberg method. Finally, all TFs with adjusted p < 0.0005 were selected to be key TFs regulating metabolic DEGs. All these procedures were implemented in R 3.4.2.

To reconstruct a sub-network describing TF-target interaction by two key TFs, we selected the target genes of the TFs and built a network model describing the key TF-target interactions and protein-protein interactions among the targets. The TF-target interactions were obtained from databases including Cistrom, mSigDB, OregAnno, CellNet and UCSC and the protein-protein interactions were obtained from BioGRID, STRING and HPRD and combined information from the three databases into one list. All these procedures were implemented in R 3.4.2. The network was displayed using Cytoscape 3.6.0.

**The analysis of genetic alterations in tumor models**. For all the cancer cell lines used for this study, the genetic alterations of driving gene mutations were annotated according to CCLE database. Mutations in PDX models were based on RNA-seq or Exome-seq analysis. Copy number alterations in PDX models were based on the microarray analysis. Wildtype NSCLC models referred to those without mutation, amplification or deletion of KRAS, EGFR, FGFR1, FGFR2, FGFR3, FGFR4, ALK, MET, PIK3CA, HER2, ROS1, RET, PDGFRA, DDR2, and PTEN genes, which are well-defined driving genes of NSCLC. For ChaGo-K-1 cell line, which was generally classified as a lung cancer cell line, additional genes including KIT, INSR, IGF1R, and MYC were analyzed for the assignment of its genotype.

**Chemicals and antibodies**. The following chemicals were used: AZD4547 (Selleck, S2801), AZD9291 (Selleck, S7297), SGX523 (Selleck, S1112), AP24534 (Selleck, S1490), Gefitinib (Selleck, S1025), UK5099 (Sigma, PZ0160), CB839 (Selleck, S7655), Etomoxir (Aladdin, E124862), Oxamate (Sigma, O2751), CBR5884 (MCE, HY-100012), NCT503 (MCE, HY-101966), GSK2837808A (MCE, 1445879) and murine IL3 (PeproTech, 213-13-10). For animal studies that required large amount of chemicals, AZD4547 (MB5756), Gefitinib (MB1112) and NCT503 (1916571-90-8) were obtained from Melone Pharmaceutical Co., Ltd. Oxamate (S123221) and Metformin (M107827) was purchased from Aladdin.

The following antibodies were used: FGFR1 (9740 S), p-EGFR Tyr1068 (3777 S), EGFR (4267 S), p-MET Tyr1003 (3135 S), MET (8198 S), p-RET Tyr905 (3221 S), RET (3223 S), p-AKT Ser473 (4060 L), AKT (4691 S), p-ERK1/2 Thr202/Tyr204 (4370 L), ERK1/2 (4695 S), p-STAT3 Tyr705 (9145 S) and STAT3 (9139 S) from Cell Signaling; p-FGFR1 Tyr653/654 (06–1433) from Millipore; PHGDH (14719–1-AP), PSPH (14513–1-AP), and GAPDH (60004–1) from Proteintech; β-actin (P30002) from Abmart. The dilution of all the primary antibody incubation in immunoblotting is 1:1000, except for the 1:5000 dilution of GAPDH and β-actin.

All the stable isotope markers were purchased from Cambridge Isotope Laboratories, including D-Glucose (U-¹³C₆, 99%, CLM-1396–10), L-Glutamine (U-¹³C₅, 99%, CLM-1822-H-0.5), Sodium Palmitate (U-¹³C₁₆, 98%, CLM-6059–1), Sodium L-Lactate (U-¹³C₃, 98%, 20% W/W in H₂O, CLM-1579-PK) and Sodium L-Lactate (3-¹³C, 98%, 20% W/W in H₂O, CLM-1578-PK).

**Immunoblotting analysis**. Cells were collected and lysed using pre-heated 2% SDS by vortexing vigorously for 2–3 s at maximum speed, followed by boiling for 30 min. Protein concentrations were determined using BCA assay (Beyotime, P0011). Proteins were subjected to SDS-PAGE, transferred to nitrocellulose membranes (Immobilon-P, Millipore) and blocked for 1 h at room temperature with 3% milk in 1xTris-buffered saline Tween-20 (TBST) (25 mM Tris, 150 mM NaCl, 2 mM KCl, pH 7.4, supplemented with 0.1% Tween-20) and blotting was performed with primary antibodies at 4 °C overnight. After washing the membranes with TBST three times for 30 min, horseradish peroxidase-conjugated anti-rabbit IgG (dilution, 1: 2000) or anti-mouse IgG (dilution, 1: 5000) antibodies were incubated at room temperature for 1 h. The membranes were washed with TBST three times for 30 min, and the antigen-antibody reaction was visualized with an enhanced chemiluminescence assay (Thermo Scientific) or Femto chemiluminescence assay (Thermo Scientific).

**Immunohistochemistry analysis**. For PC9 xenograft and LU-01-0251 PDX models, tumor samples were collected at 6 h after the last dosing. For the rest of PDX models, including EGFR mutant tumors (n = 6), FGFR amplified tumors (n = 4) or wildtype tumors (n = 6), used for immunohistochemistry analysis, basal level of metabolic enzyme genes was detected. Three independent tumor tissues from each PDX model were analyzed. The tumor tissues were fixed with

paraformaldehyde (4%) before dehydration and paraffin embedding. Paraffin sections were stained with hematoxylin and eosin (H&E) according to standard protocols or were subjected to immunohistochemical staining using a horseradish peroxidase-labeled streptavidin-biotin ABC kit (ZSGBBIO, Beijing, China) with hematoxylin as the counterstain. Anti-EGFR, anti-p-EGFR Tyr1068, anti-ATF4, anti-cMyc, anti-PHGDH, anti-PSPH, an-HK2, anti-LDHA and Ki67 were diluted at the recommended ratio according to the manufacturer's instructions in 0.1% BSA/PBS and incubated on slides in a humidified chamber for 2 h, and imaged at × 100 magnification.

**GC-MS analysis.** Cell lines were cultured under indicated conditions to identify metabolic characteristics. In $[U-^{13}C_6]$-glucose and $[U-^{13}C_5]$-glutamine tracer experiments, BAF3 and BAF3-RTK cells were cultured in the DMEM medium (Sigma-Aldrich, D5030) by adding 10 mM $[U-^{13}C_6]$-glucose, 4 mM label-free glutamine or 10 mM label-free glucose and 4 mM $[U-^{13}C_5]$-glutamine for indicated hours. In $[U-^{13}C_{16}]$ palmitate tracer experiments, $[U-^{13}C_{16}]$ palmitate was first non-covalently conjugated to fatty acid free BSA (Applygen, A2000) and then the 100 μM BSA-conjugated $[U-^{13}C_{16}]$ palmitate and 1 mM carnitine were added to culture medium for fatty acid oxidation assay. In the competitive uptake experiment of glucose and lactate, BAF3-EGFR and BAF3-FGFR cells were cultured in the presence of 5 mM $[U-^{13}C_3]$-lactate and 10 mM label-free glucose for indicated hours. In the analysis of $[U-^{13}C_6]$-glucose-derived serine, PC9 cells were transfected with indicated siRNA for 48 h followed by 24h-culture in the DMEM medium (Sigma-Aldrich, D5030) by adding 10 mM $[U-^{13}C_6]$-glucose and 4 mM label-free glutamine. The $^{13}C$-labeled fraction contribution of the metabolite isotopologues was analyzed by GC-MS and the raw MS data was corrected for the contribution of all naturally abundant isotopes. The metabolite intensities in the metabolomics data of BAF3 and BAF3-RTK cells were analyzed by GC-MS. In the analysis of intracellular lactate and pyruvate level, RT112 cells were transfected with indicated siRNA for 48 h and the metabolite intensities were analyzed by GC-MS.

Sample preparation: The cell samples were collected ($5–7 × 10^6$ cells per sample) in 2-mL Eppendorf tubes with 1.6 mL of 80% (v/v) methanol solution for each sample. Then, the sample was processed by 7 cycles of 2 min ultra-sonication and 2 min interval in ice-bath. The cell lysates were kept at 4 °C for 1 h prior to centrifugation at 14,000 × g and 4 °C for 15 min. The supernatant was evaporated to dryness with the vacuum centrifuge.

Derivatization: The dried sample for metabolomics analysis was reconstituted in 30 μL of 20 mg/mL methoxyamine hydrochloride in anhydrous pyridine, and incubated at 37 °C for 90 min. Following the supplementation of another 30 μL of N,O-bis(trimethyl)trifluoroacetamide (BSTFA) with 1% trimethylchlorosilane (TMCS), the sample was derivatized at 70 °C for 60 min prior to GC-MS metabolomics analysis. The dried sample for isotopologue spectral analysis (ISA) was reconstituted in 30 μL of 20 mg/mL methoxyamine hydrochloride in anhydrous pyridine, and incubated at 37 °C for 90 min. Samples were then derivatized by addition of 30 μL of N-tert-butyldimethylsily-N-methyltrifluoroacetade (MTBSTFA) with 1% tert-butyldimethylsilyl (tBDMS) (Regis Technologies) and incubated at 55 °C for 60 min.

Instrumental Analysis: Instrumental analysis was performed on an Agilent Intuvo 9000 gas chromatography system coupled to an Agilent 5977B inert MSD system (Agilent Technologies Inc, CA, USA). A HP-5MS fused-silica capillary column (30 m × 0.25 mm × 0.25 μm; Agilent J&W Scientific, Folsom, CA) was utilized to separate the derivatives. Helium ( > 99.999%) was used as a carrier gas at a constant flow rate of 1 mL/min through the column. Injection volume was 1 μL in the splitless mode, and the solvent delay time was set 6 min. The initial oven temperature was held at 70 °C for 2 min, ramped to 160 °C at a rate of 6 °C/min, to 240 °C at a rate of 10 °C/min, to 300 °C at a rate of 20 °C/min, and finally held at 300 °C for 6 min. For ISA, injection volume was 1 μL in the splitless mode, and the solvent delay time was 5.5 min. The initial oven temperature was held at 100 °C for 2 min, ramped to 180 °C at a rate of 10 °C/min, to 260 °C at a rate of 5 °C/min, to 300 °C at a rate of 10 °C/min and finally held for 8 min. The temperatures of injector, transfer line, and electron impact ion source were set 250 °C, 250 °C, and 230 °C, respectively. The electron energy was 70 eV, and data was collected in a full scan mode (m/z 50–600).

**UHPLC-QTOF-MS analysis.** BAF3 and BAF3-RTK cells were cultured in the DMEM medium (Sigma-Aldrich, D5030) by adding 10 mM $[U-^{13}C_6]$-glucose and 4 mM label-free glutamine for 24 h. The incorporation percentage of $[U-^{13}C_6]$-glucose to purine nucleotides and glutathione (GSH) was analyzed by UHPLC-QTOF-MS. In the analysis of $[U-^{13}C_6]$-glucose-derived purine nucleotides, PC9 cells were transfected with indicated siRNA for 48 h followed by 24 h-culture in the DMEM medium (Sigma-Aldrich, D5030) by adding 10 mM $[U-^{13}C_6]$-glucose and 4 mM label-free glutamine. The incorporation percentage of $[U-^{13}C_6]$-glucose to purine nucleotides was analyzed by UHPLC-QTOF-MS. The metabolite intensities in the metabolomics data of BAF3 and BAF3-RTK cells were analyzed by GC/MS and UHPLC-QTOF-MS.

The metabolite extraction process was same as the GC–MS analysis section. The dry residues of metabolite extracts were reconstituted with 200 μL 50% (v/v) acetonitrile solution and kept at −20 °C for 30 min. Then vortexed for 30 s before centrifugation at 12,000 × g for 10 min at 4 °C, and 3 μL of the supernatants were injected into the LC-MS for measurement. Quality control (QC) sample was the

mixture prepared from an equal amount in each sample supernatant and analyzed with the same procedure as that for the experiment samples.

For reversed-phase liquid chromatography, Agilent uHPLC (Binary Pump G7120A, Multisampler G7167B and Column Comp G7116B) was employed with chromatographic separation on an Agilent rapid resolution HD C18 column (1.8 μm, 3.0 × 150 mm internal dimensions, PN:959759–302). The column was maintained at 30 °C and the injection volume of all samples was 3 μL. The mobile phase consisted of 0.1% formic acid in LC-MS grade water (mobile phase A) and LC–MS grade acetonitrile (mobile phase B) run at a flow rate of 0.4 mL/min. The analysts were separated with following gradient program: 2% B held for 1 min, increased to 40% B in 4 min, increased to 70% B in 7 min, increased to 95% B in 3 min, held for 5 min, and the post time was set 6 min. The mass spectrometer (Agilent MS QTOF 6545 A) was operated in positive ion mode with a 3.5 kV capillary voltage. Nozzle voltage 500 v for positive ion mode, 1000 v for negative ion mode. The source gas and sheath gas temperature were set at 200 °C and 325 °C, respectively.

For nucleotide analysis, Agilent uHPLC was employed with chromatographic separation on a Thermo hypercarb column (3 μm, 2.1 × 150 mm internal dimensions, PN: 35003–152130). The column was maintained at 35 °C and the injection volume of all samples was 3 μL. The mobile phase consisted of 20 mM ammonium acetate and 3 mL/L ammonium hydroxide (>28%) in LC-MS grade water (mobile phase A) and 20 mM ammonium acetate and 3 mL/L ammonium hydroxide (>28%) in LC–MS grade 90% acetonitrile (v/v) (mobile phase B) run at a flow rate of 0.3 mL/min. The analysts were separated with the following gradient program: 3% B held for 2 min, increased to 45% B in 9 min, increased to 95% B in 4 min, held for 2 min, and the post time was set 5 min. The mass spectrometer was operated in positive ion mode with a 3.5 kV capillary voltage. Nozzle voltage was set 500 V for positive and 1000 V for negative ionization mode. The source gas and sheath gas temperature was set 200 °C and 325 °C, respectively.

**RNA isolation and RT-qPCR analysis.** Total RNA was extracted with TRIzol reagent (Thermo Fisher Scientific, 15596026) and subjected to reverse transcription with PrimeScript® RT reagent Kit (Takara, RR014A). PCR reactions were performed with SYBR® Premix Ex Taq™ kit (Takara, RR420A) using ABI Prism VIIA7 Real-Time PCR System. All measurements were performed in duplicate and the arithmetic mean of the Ct-values was used for calculations: target gene mean Ct-values were normalized to the respective housekeeping genes (GAPDH or β-Actin), mean Ct-values (internal reference gene, Ct), and then to the experimental control. Obtained values were exponentiated 2-ΔΔCt to be expressed as n-fold changes in regulation compared with the experimental control (2-ΔΔCt by the method of relative quantification. The assay was performed in biological triplicates, and error bars represented SD.

**siRNA transfection.** siRNA transfection in cancer cells was performed using Lipofectamine RNAiMAX Reagent Agent (Invitrogen, 13778150) in OPTI-MEM serum-free medium (Gibco, 31985–070) according to the manufacturer's instructions.

The siRNAs were synthesized as RP-HPLC-purified duplexes by GenePharma (Shanghai, China), dissolved in DEPC water at the initial concentration of 20 μM and preserved at −20 °C. Scrambled siRNA was used as a negative control (NC).

**Cell proliferation assay.** For CCK8 assay (Vazyme, 606051), cells were seeded in 96-well plates for overnight and then treated with DMSO or the indicated inhibitors for 72 h. Untreated cells served as the indicator of 100% cell viability. The absorbance (optical density, OD) was read at a wavelength of 450 nm on an ELISA plate reader. For sulforhodamine B (SRB) assay (Sigma-Aldrich, S1402), cells were seeded in 96-well plates at density of $3–5 × 10^3$ cells per well for overnight and then were treated with DMSO or the indicated inhibitors for 72 h. The culture medium was aspirated, and 10% trichloroacetic acid (TCA) was added to each well and allowed to stand for 24 h at 4 °C to precipitate proteins. Then, the precipitated proteins were stained for 15 min at room temperature with 0.4% (w/v) SRB in an acetic acid solution 1% (v/v), washed with 1% acetic acid 5 times and then dried. The adherent SRB was solubilised in 10 mM Tris buffer and the absorbance was read at a wavelength of 560 nm on an ELISA plate reader. Cell viability rate was calculated as follows: ($OD_{treated}/OD_{control}$) × 100%. The assay was performed in biological triplicates, and error bars represented SD.

For the cells in Figs 2c and 3k, the adherent cells were seeded in 12-well plates per well for overnight and then were treated with DMSO or the indicated inhibitors for 6 days. Then the culture medium was aspirated, and fixed with 90% ethanol for 30 min, followed by the staining with 0.1% crystal violet (Sigma-Aldrich, C0775). Plates were air dried overnight. Crystal violet was eluted with 30% (v/v) acetic acid solution and the absorbance was read at a wavelength of 600 nm on an ELISA plate reader. The non-adherent cells were seeded in 12-well plates per well for overnight and then were treated with DMSO or the indicated inhibitors, followed by cell number counting after 6 days. Cell growth inhibition rate was calculated as follows: ($1-OD_{treated}/OD_{control}$) × 100%. The assay was performed in biological duplicates.

**Seahorse XF analysis.** Seahorse XF96 assay well was equipped with a disposable sensor cartridge and embedded with 96 pairs of fluorescent biosensors (oxygen and pH), coupled to fiber-optic waveguides (Seahorse Bioscience). The XF Calibrant

Solution 200 μL/well was added into the sensor box and put into the pre-work station at 37 °C (non-CO$_2$ incubator) for hydration overnight. To measure oxygen consumption rate (OCR) and extracellular acidification rate (ECAR), BAF3 and BAF3-RTK cells ($2 \times 10^6$ per well) and cancer cells ($1-1.5 \times 10^5$ per well) were plated into XF96 cell culture plates in 200 μL XF-Base Medium Minimal DMEM supplemented with glucose (25 mM) and pyruvate (1 mM). The cells were incubated for 1 h at 37 °C (non-CO$_2$ incubator) for pH stabilization. The measurement of oxygen consumption was expressed in pmol/min and extracellular acidification rate was expressed in mPH/min. The experiment was repeated for three times independently and the shown was a representative result. The shown data were means of biological triplicates and error bars represented SD.

**Lactate measurement**. Lactate levels were measured using a Lactate Colorimetric/Fluorometric Assay Kit (Biovision, K607–100). Cells were seeded in 12-well plates overnight and then were harvested to collect both of the supernatant and pellet for lactate release and intracellular lactate assay respectively. Cell pellet was lysed in the 100 μL buffer provided by the kit and centrifuged at $12,000 \times g$ for 15 min to collect the cell supernatant. The resultant supernatant of both fractions was mixed with the assay solution. The absorbance was measured at 570 nm and the readout was normalized by the protein amounts. The experiment was repeated for three times independently and the shown was a representative result. The shown data was performed in biological triplicates, and error bars represented SD.

**Intracellular ATP measurement**. Intracellular ATP levels were assessed using an ATP assay kit (Beyotime, S0027). Cells were lysed in 100 μL buffer provided by the kit and centrifuged at $12,000 \times g$ for 15 min to collect the cell supernatant. An aliquot of ATP detection working solution was added to a 96-well culture plate and was incubated for 5 min at room temperature. Then, the cell lysate was added to the wells, and the luminescence was measured immediately. The readout was normalized by the protein amounts of each well. The experiment was repeated for three times independently. The shown data was performed in biological triplicates, and error bars represented SD.

**ROS measurement**. The levels of ROS were detected using 2′,7′-Dichlorofluoresci diacetate (DCFH) (Sigma, D6883). Cells were incubated with 10 μM DCFH at 37 °C for 30 min. During the incubation period, each sample was agitated every 10 min to ensure that the reagent reacted sufficiently with the ROS. Then the cells were centrifuged at $500 \times g$ for 5 min to collect the cell pellet, followed by PBS rinse twice to reduce the fluorescence background. The level of ROS was measured by FACS analysis (FACSCalibur flow cytometer, BD Biosciences). The experiment was repeated for three times independently and shown was a representative result. The shown data was performed in biological triplicates, and error bars represented SD.

**Glucose/glutamine dependency analysis**. Cells were seeded in 96-well plates per well for overnight and then change medium by RPMI-1640 medium at the indicated concentration of glucose (GLC) or glutamine (GLN). Briefly, for glucose dependency assay, cells were cultured in the RPMI-1640 medium with no glucose (Gibco, 11879020) or in the medium containing 4 mM glucose. For glutamine dependency assay, cells were cultured in the RPMI-1640 medium (no glutamine) (Gibco, 21870076) containing 4 mM or 0.5 mM glutamine. Subsequently, the cell culture plates were placed into the live-cell analysis system IncuCyte® (Essen BioScience Ltd., Hertfordshire, UK), and images were acquired every 6 h by automated real-time assessment for 4 days. Growth curves were plotted as the change in confluence percentage. In the glucose/glutamine dependency analysis of BAF3 and BAF3-RTK cells, cells were cultured in the RPMI-1640 medium (no glucose) (Gibco, 11879020) with or without adding glucose (4.5 mM), and in the RPMI-1640 medium (no glutamine) (Gibco, 21870076) with or without adding glutamine (4.5 mM) respectively. Growth curves were plotted by measuring growth fold change by counting cell numbers at indicated time.

**Statistics**. Statistical significance was analyzed using two-tailed Student's $t$ test or Fisher's exact test, and $p < 0.05$ was considered to be statistically significant.

**Reporting summary**. Further information on research design is available in the Nature Research Reporting Summary linked to this article.

## Data availability

All data presented in the main text and the supplementary information are available. The transcriptome data of BAF3 and BAF3-RTK cells have been deposited in the GEO under ID code GSE111292. The source data underlying all Figs. and Supplementary Figs. are provided as a Source Data file.

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

### Acknowledgements

This work was supported by the China International Science and Technology Cooperation Program (No. 2015DFM30040 to M.H.), the National Key Research and Development Program of China (No. 2016YFC0904800 to S.H.L.), the National Natural Science Foundation of China (No. 81821005 to M.G., No. 81672911 to S.H.L., No. 81573464 to M.H.), the Strategic Priority Research Program of the Chinese Academy of Sciences (No. XDA12000000 to M.G), and the K. C. Wong Education Foundation.

### Author contributions

M.H., S.H.L and N.J. conceived the project; M.G. and J.D. provided supervision; N.J., A.B., X.L. performed the research and analyzed the data; X.W., Y.L., T.W., Z.C. performed part of the experiments; H.Z. J.X. and M.T. provided assistance in data analysis; J.A., H.X. and T.Z. generated BAF3-RTK cell lines; S.T., D.L., R.H., E.L.L. and X.Y. provided technique assistance; Y.S. assisted mass spectrometry analysis; M.H., S.H.L. and N.J. wrote the manuscript. All authors approved the final version of the manuscript.

### Additional information

**Competing interests:** The authors declare no competing interests.

