## [Peer Review File · Nature Communications]

Reviewers' Comments:

Reviewer #1:

Remarks to the Author:

This manuscript (by Jin et al) described the effect of RTK activation on metabolic vulnerability. The authors used Baf3 cells transfected with activated RTK, human cancer cell lines, PDX and data from TCGA database to analyze the effects, including metabolic profiling, gene expression correlation, and cancer cell proliferation in vitro and in vivo. Overall, the finding is novel, and potentially interesting for publication. However, there are several issues need to be clarified;
Comments

1. Correlation analysis is mainly based on TCGA database, which is mostly from NGS profiling. Validation with alternative approaches, such as qPCR, IHC or Western blot analysis on clinical specimens will strengthen implications.
2. The results from TCGA data (Fig 2k, 3i) there were substantial overlap on gene expression between subgroups. P values were not impressive based on large set of data. What is FDR? Were there any other metabolic genes different more significantly?
3. There are several FGFR genes, which genes were included in the study in BAF3, PDX, cell lines etc should be specified. Does each FGFR gene have similar metabolic effects?
4. Information for PDX models should be detailed, including mutations in RTK and other cancer related genes. What was the effect of PDXs with wt RTK?

Specific comments:

1. Figure 1a and table S1, it is not clear what the bar and values indicate, the IC50 or % of cell viability? Overall the difference were very mild.
2. Fig 1b, image is not clear, either move to Supplement or make it clear, including legend. Are there any subgroups in EGFR, FGFR and RET mutations?
3. Fig 1i and table S5, Enrichment tables in S5 were difficulty to read. What criteria were used for Fig 1i, based on p value, enrichment score, FDR?
4. Fig 2b, Dose response should be determined and IC50 calculated. The difference based on one dose was very mild.
5. Fig 2m, the co-amplification/deletion for EGFR and PSPH is likely caused by physical linkage of the two genes in chr 7p. Check other genes nearby may lead to the similar results. It does not prove that two genes have functional interactions.
6. Fig 3h: what were results from human cancer cell lines and clinical specimens, similar changes as in Baf3 cells?

Reviewer #2:

Remarks to the Author:

REVIEW

Jin et al report a dissection of metabolic vulnerabilities associated with specific receptor tyrosine kinase (RTK) alterations encountered in tumors. They modeled different RTK effects in the context of a common cell line (BAF3). This analysis has led them to the identification of differential metabolic pathway utilization in the context of aberrant EGFR and FGFR. In the first case, the cells relied more heavily on the serine biosynthetic pathway. In the second case, the cells relied more heavily on glycolysis. The authors then extrapolate their findings beyond the BAF3 model to cancer cell lines endogenously harboring some of the oncogenic RTK alterations. Finally, authors elucidate the network of transcription factors controlling the differential metabolism observed. Collectively, this work aims to identify novel nodes for therapeutic intervention in specific cancer subsets.

While the authors choose to address an important question in the field of oncology and tumor metabolism, there are MULTIPLE SIGNIFICANT concerns that would have to be addressed before the manuscript would be considered ready for publication.

Major comments:

- Metabolic changes in cancer cell lines are heavily dependent on cell culturing conditions. It is currently not clear from the manuscript to what extent some of the changes are driven by the media differences vs oncogenes encountered in the cell lines. The BAF3 experiments using isogenic cell lines are internally controlled. However, comparisons across different cancer cell lines should be done in the same media to enable direct comparisons.
- It's not clear whether the BAF3 experiments were done in the presence or absence of IL3. It is important to annotate this clearly. To what extent would the answer be different in the +/- IL3 conditions?
- Figure 1b. The expression analysis of TCGA samples in superficial. Given that some of the tumor genotypes cluster by tissue type (e.g., EGFR mutations are encountered in lung cancer, but not in AML), it's not clear that metabolic clusters presented are not just a reflection of tissue types. Therefore, similar analysis of KEGG enzymes should be done within a particular cancer type, thus accounting for the confounding factor of tissue of origin. Any tumor genotype associations identified in such a way, would be significantly more meaningful.
- The authors do a nice job of studying particular genetic lesions using the BAF3 model system. However, the follow up in endogenous cell lines panels is not sufficiently extensive. Therefore, the case for specificity of vulnerabilities associated with each of the genotypes is somewhat weak. One way to overcome this is to increase the number of cell lines analyzed. Alternatively, is there additional information that can be gleaned from publicly available datasets, particularly ones focusing on tumor dependence (Project Achilles, Project DRIVE, depmap, etc.)?
- The in vivo experiment shown in Figure 3d is potentially nice. However, it's hard to interpret without seeing the fractional enrichment in the blood, as only tumor results are shown.
- In Figure 3k the labeling results are potentially quite nice. However, showing citrate+2 results would significantly strengthen the authors' case. The choice of citrate+6 is peculiar, given the fact that it's likely to be found in exceedingly small amounts in the tumors and very close to the background noise.
- While oxamate is a reasonable starting tool compound for inhibiting lactate dehydrogenase, it lacks specificity (PMID: 27538376). Therefore, at least in vitro, some additional inhibitors should be used to support authors' claims. Fortunately, multiple specific lactate dehydrogenase inhibitors have been published to date (PMID: 27479743, 22417091, 23302067, 24280423)
- The connection of ATF4 transcription factor regulating the serine biosynthetic pathway is not novel (PMID: 26482881) and the authors don't cite it. Furthermore, authors need to account for the NRF2 pathway connection (via ATF4 activation) to the serine pathway. This suggestion pertains to bioinformatic analyses (accounting for EGFR mutations vs NRF2 pathway activation mutations) and to the choice of cell lines used for follow-up studies.

Minor comments:

- Figure 1c. The axis is labeled "proliferation curve (relative)". A more accurate description should be something along the lines of "relative cell number". Similar comment applies to all figures displaying relative rates of cell growth.
- There are several figure panels that are completely not discussed in the text of the manuscript (Figure 2d, Supplemental Figure 2f, etc.). It's worth adding some text to describe them.
- The authors are very strong in some of their conclusions, even in the context of sparse data. It's strongly recommended that they adopt a more conservative language to state their potential conclusions. For example, line 126 ("...faithfully echoed..."). For example, line 137 (the text states that CB839 only affected RET-aberrant cells). Certainly, RET cells were the most impacted, but other cell lines showed significant effects as well. For example, line 261 ("... proving our findings in vivo.").
- Only one of the endogenous cell lines harboring RET alteration displayed differential growth upon glutamine withdrawal (TT), while the other cell line (TPC-1) was not particularly affected. This data is shown in Supplemental Figure 1i, but the text in the manuscript overstates the case by saying that all RET lines were affected.
- Figure 1i. Metabolic pathway enrichment is shown here as function of the BAF3 genotype. However, it would also be useful to see all pathways that are enriched, and not restricting the

analysis to metabolism only. This will give a reader a better idea whether metabolic rewiring is a major component of what is happening downstream of these RTKs or not.

- Supplementary Figure 2a is not discussed in the text. The labels in the left panels are likely incorrect. Presumably, they are meant to compare BAF-EGFR vs BAF-FGFR.
- The manuscript text overstates the case of specificity of PHGDH and PSPH effects on EGFR-aberrant cell lines. Supplementary Figure 2g clearly shows that 1 out of the 2 FGFR-aberrant cell lines also is affected by PHGDH and PSPH knockdown. Does the left most panel in that figure contain a correct label or correct control? Right now "EGFR" is used as a control here, but FGFR would make more sense as a control.
- PHGDH inhibitor used, CBR5584, is not cited in the bibliography.
- Line 198. The discussion of PSPH being a rate-limiting enzyme of the serine biosynthetic pathway is incomplete. While this enzyme can be the rate-limiting in the context of the liver, it may not be in the context of cancer. Please see PMID: 21760589.
- The relevance of the findings shown in Figure 2l is not particularly clear. Similarly, the analysis shown in Figure 2m is not clearly described. What are the cutoffs used for defining PSPH amplified tumors? Is it a focal amplification, similar to the one encountered with PHGDH? Alternatively, is it a broad chromosome-wide copy number gain (in which case it would be harder to make a case for PSPH contribution)? Please provide some statistical analysis that these two co-occurring events (EGFR mutation and PSPH amplification) are significant, beyond what would be expected by chance.
- Figure 3c label "Lactate relative absolute intensity" is not clear. Please explain.
- Please provide more details on how the amplifications were defined in Figure 3i.
- Line 269. Instead of the phrase "...mitochondrial phosphorylation..." it should read "...mitochondrial respiration...".

Point-to-point responses to reviewers' comments

Reviewer #1 (Remarks to the Author): Expert in lung cancer

This manuscript (by Jin et al) described the effect of RTK activation on metabolic vulnerability. The authors used Baf3 cells transfected with activated RTK, human cancer cell lines, PDX and data from TCGA database to analyze the effects, including metabolic profiling, gene expression correlation, and cancer cell proliferation in vitro and in vivo. Overall, the finding is novel, and potentially interesting for publication. However, there are several issues need to be clarified.

Response: We thank the reviewer for the comment that “**the finding is novel and potentially interesting for publication**”. We are also grateful for the valuable advice from the reviewer, which have helped us greatly improve the manuscript.

1. Correlation analysis is mainly based on TCGA database, which is mostly from NGS profiling. Validation with alternative approaches, such as qPCR, IHC or Western blot analysis on clinical specimens will strengthen implications.

Response: As the reviewer advised, we have validated the TCGA data-based findings using IHC analysis of a panel of patient-derived xenograft tumor samples, which have been shown to conserve the clinical features of individual patients during limited passage in mice¹.

By comparing the NSCLC tumors with mutant (n=6) or wildtype (n=6) *EGFR*, we confirmed the upregulation of serine synthesis enzymes, such as PHGDH and PSPH, in *EGFR*-mutant tumors. Likewise, the increased expression of glycolytic enzymes, in particular HK2 and LDHA, were upregulated in *FGFR1/2*-amplified tumors (n = 4) (see figure i below).

These results, which have been provided in the revised manuscript (Fig. 3j, Supplemental Fig. 2h, 3d), strengthened our findings discovered by the analysis of the TCGA database.

Figure i. Immunohistochemical analysis of metabolic enzymes in tumors derived from PDX models. **(a)** PHGDH and PSPH staining in tumors with wildtype or mutant *EGFR*; **(b)** HK2 and LDHA staining in tumors with wildtype or amplified *FGFR1/2*.

2. The results from TCGA data (Fig 2k, 3i) there were substantial overlap on gene expression between subgroups. P values were not impressive based on large set of data. What is FDR? Were there any other metabolic genes different more significantly?

Response: We thank the reviewer for the valuable comment, which helped us look into these data more carefully.

1) The reviewer is correct that the P values were not impressive, which may be partially due to the large dataset mixed with different tumor tissue types. In the revised manuscript, we specifically extracted the lung adenocarcinoma data (n = 740). The expression of glycolytic genes in *FGFR* amplified patients and serine synthesis pathway gene expression in *EGFR* mutant tumors were significantly increased, as supported by both p values and FDR values (Fig. 2j and 3i).

2) Responding to reviewer's second question, we did discover quite a few other KEGG annotated metabolic genes that changed more significantly in *FGFR* or *EGFR* activated tumors (shown below). However, most of genes are not associated with related pathway changes or the subsequent functional outcomes.

Figure ii. Metabolic genes expression in 740 lung adenocarcinoma patients extracted from the TCGA database. **(a)** Glycolytic genes upregulated in *FGFR1/2* amplified tumors; **(b)** Genes in serine synthesis pathway increased in *EGFR* mutant patients; **(c)** Other metabolic genes annotated by KEGG increased in *FGFR* or *EGFR* aberrant tumors.

3. There are several *FGFR* genes, which genes were included in the study in *BAF3*, *PDX*, cell lines etc should be specified. Does each *FGFR* gene have similar metabolic effects?

Response: In the revised manuscript, we have specified the information of *FGFR* isoforms and provided sets of new data to carefully compare the cellular phenotypes resulted from *FGFRs* alteration. Our data support the similar metabolic effects driven by different *FGFR* isoforms.

1) *FGFR*-driven tumor models used in the study cover *FGFR1* amplification, *FGFR2* amplification and *FGFR3* mutation/amplification/fusion (see Table i below). We have specified this information in the revised manuscript (Supplementary Table 5 and Table 7).

Table i. *FGFR* gene status in tumor models used in this study

Models	Name	Gene alterations	Models	Name	Gene alterations
Cell lines	NCI-H520	FGFR1 amplification	PDXs	LU0894	FGFR3 amplification
	NCI-H1581	FGFR1 amplification		LU6416	FGFR2 amplification
	DMS114	FGFR1 amplification		LU1302	FGFR2,3 amplification
	NCI-H2444	FGFR1 amplification		LU1901	FGFR1 amplification
	NCI-H716	FGFR2 amplification		LU2504	FGFR2,3 amplification

SNU16	FGFR2 amplification	LU6408	FGFR1 amplification
KATO III	FGFR2 amplification	LU6429	FGFR2 amplification
RT112	FGFR3 amplification, FGFR3-TACC3 fusion	LU0743	FGFR2 amplification
UM-UC-14	FGFR3 S249C mutation		

2) These tumor models, regardless of different FGFR gene alterations, showed similar metabolic phenotypes, including glucose dependency (Supplementary Fig. 1k), the response to LDH inhibition (Fig. 3k) and upregulated expression of glycolytic enzymes (Fig. 3j, Supplementary Fig. 3d).

3) We have also compared the oncogenic forms of FGFR1 (*TEL-FGFR1*) and FGFR3 (*TEL-FGFR3*) in the same BAF3 cell context. Both genes resulted in the IL3-independent cell growth, the exquisite sensitivity to specific FGFR inhibitors and the enhancement of both aerobic glycolysis and oxidative phosphorylation, as indicated by ECAR and OCR, compared to BAF3 parental cells. These results have been included in the revised manuscript (Supplementary Fig. 1d-1f).

Figure iii. The comparison of BAF3-FGFR1 (*TEL-FGFR1*) and FGFR3 (*TEL-FGFR3*) cells. **(a)** Cell growth in the presence/absence of IL3; **(b)** Cell sensitivity to a pan-FGFR inhibitor AZD4547; **(c)** OCR and ECAR determined by Seahorse XF96 analyzer.

4. Information for PDX models should be detailed, including mutations in RTK and other cancer related genes. What was the effect of PDXs with wt RTK?

Response: The detailed information of PDX have been provided in the revised manuscript (Supplementary Table 7). PDX models and cancer cells lines without detectable driving gene alterations were tested as suggested by the reviewer. According to our new results, wildtype tumor,

though appearing heterogeneous, are mostly nonresponsive to inhibitors targeting serine synthesis or lactate production.

1) In the revised manuscript, a total of 21 PDX models were used for different assays. The detailed information, including the RTK mutations and other oncogenes has been provided in Supplementary Table 7. Part of information are also shown below (Table ii).

Table ii. RTK genetic alterations in PDX models

Name	RTK alterations	Cancer type	Name	RTK alterations	Cancer type
LU1901	FGFR1 amplification	NSCLC	LU6422	EGFR L858R	NSCLC
LU6408	FGFR1 amplification	NSCLC	LU0858	EGFR L858R	NSCLC
LU1302	FGFR1 amplification	NSCLC	LU-01-0251	EGFR L858R	NSCLC
LU2504	FGFR1,2 amplification	NSCLC	LU1868	EGFR L858R/T790M	NSCLC
LU6429	FGFR2 amplification	NSCLC	LU1235	EGFR E746_A750del	NSCLC
LU0743	FGFR2 amplification	NSCLC	LU5205	EGFR L747_S752del	NSCLC
			LU0861	EGFR L747_P753del	NSCLC
LU6411	WT	NSCLC	LU6471	WT	NSCLC
LU2071	WT	NSCLC	LU-01-0393	WT	NSCLC
LU6401	WT	NSCLC	LU-01-0416	WT	NSCLC
LU6416	WT	NSCLC	LU0894	WT	NSCLC

2) Upon the reviewer's request, we have tested the response of PHGDH and LDH inhibitors in three randomly picked NSCLC PDX models without detectable driving gene alterations (LU-01-0393, LU-01-0416 and LU2071). None of these models showed the significant response to these inhibitors (Supplementary Fig. 3k).

Figure iv. Response of PHGDH and LDH inhibitors in PDX models. Mice carrying indicated tumors were dosed daily with NCT503 (40 mg/kg) or Oxamate (750 mg/kg) for indicated days. Tumor volume were measured every 3 days.

3) We also tested the PHGDH and LDH inhibitors in an epidermoid carcinoma cell line A431 xenograft, a widely used control cell line in RTK-related studies². This model also barely responded to the treatment.

Figure v. Response of PHGDH and LDH inhibitors in A431 xenograft model. Mice carrying indicated tumors were dosed daily with NCT503 (40 mg/kg) or Oxamate (750 mg/kg) for indicated days. Tumor volume were measured every 3 days.

Specific comments:

1. Figure 1a and table S1, it is not clear what the bar and values indicate, the IC50 or % of cell viability? Overall the difference was very mild.

Response: In Fig. 1a and Table S1 (Supplementary Fig. 1a and Supplementary Table 1 in the revised manuscript), the bars indicate the inhibitory rate of cell growth upon the treatment of the indicated metabolic inhibitors. This figure represents the unbiased hierarchical cluster analysis of the indicated cell lines according to their response to various metabolic inhibitors. FGFR and EGFR aberrant cells are present in two obvious clusters. We have revised the figure and the description to make it easier to understand in revised manuscript.

We agree that the difference was relatively mild due to the limited cell numbers in each genotype, and this figure has been moved the Supplementary Fig. 1a.

2. Fig 1b, image is not clear, either move to Supplement or make it clear, including legend. Are there any subgroups in EGFR, FGFR and RET mutations?

Response: We have moved this figure to Supplementary Fig. 1b as suggested.

Also, to eliminate the bias from different tissue types, we specifically extracted 740 lung adenocarcinoma data and re-analyzed the results. We found that *EGFR* activation mutation (n = 25), *FGFR1/2* amplification (n = 15) and *RET* gene fusion (n = 2) are better clustered in metabolic gene expression pattern (See Figure vi below). Further looking into the KEGG pathways in these samples discovered that serine and related metabolism pathways were upregulated in *EGFR* mutated cancer patients (n = 25) versus the counterpart with wildtype *EGFR* (n = 715) (Fig. 2a). However, there is no obvious subgroups of metabolic gene expression in *EGFR*, *FGFR* and *RET* activation samples.

Figure vi. Heatmap obtained by cluster analysis using the mRNA data of metabolic genes of 740 lung adenocarcinoma patients in the TCGA data sets.

3. Fig 1i and table S5, Enrichment tables in S5 were difficult to read. What criteria were used for Fig 1i, based on p value, enrichment score, FDR?

Response: Enrichment tables in Supplementary Table 5 (Supplementary Table 6 in the revised manuscript) has been improved. Fig. 1i is the KEGG pathway enrichment analysis by the differentially transcribed clusters showing up in the RNA-seq in Supplementary Fig. 1o, in which the different clusters were color-coded. The criteria for Fig. 1i are p value.

Figure vii. Transcriptome analysis of BAF3-RTK cells. Left panel: Heatmap of transcriptome profiling representing the mRNA levels of genes performed by RNA-seq. The rows indicated different genes, and the columns indicated different cells (n = 3 per cell line). Right panel: KEGG pathway enrichment analysis of the differentially transcribed clusters color-coded in the heatmap of the left panel.

4. Fig 2b, Dose response should be determined and IC50 calculated. The difference based on one dose was very mild.

Response: We thank the reviewer for this important point. In the revised manuscript, we have greatly increased the number of cell lines and tested 3 different doses of PHGDH inhibitors. A total of 24 cancer cell lines, including 8 *FGFR1/2/3* amplified/mutant cells, 8 *EGFR* mutant cells and 8 wildtype cells lacking detectable driving gene alterations were tested (See the detailed information of the cell lines in Supplementary Table 5).

The results showed that *EGFR*-mutant cancer cells showed preferential sensitivity to PHGDH inhibition compared with *FGFR*-altered cancer cells or wildtype cancer cells. These new results have been included in the revised manuscript (Fig 2c, also see Figure viii blow).

a

Name	Gene alterations	Name	Gene alterations	Name	Gene alterations
NCI-H520	FGFR1 amplification	PC9	EGFR delE746-A750	A431	Unknown
NCI-H1581	FGFR1 amplification	HCC827	EGFR delE746-A750	NCI-H1299	Unknown
NCI-H2444	FGFR1 amplification	HCC4006	EGFR delE746-A750	NCI-H2170	Unknown
NCI-H716	FGFR2 amplification	NCI-H1650	EGFR delE746-A750	ABC-1	Unknown
SNU-16	FGFR2 amplification	HCC2279	EGFR delE746-A750	ChaGo-K-1	Unknown
KATO III	FGFR2 amplification	NCI-H1975	EGFR L858R , T790M	NCI-H226	Unknown
RT112	FGFR3 amplification, FGFR3-TACC3 fusion	NCI-H2172	EGFR L858R	SK-MES-1	Unknown
UM-UC-14	FGFR3 S249C mutation	NCI-H3255	EGFR L858R	NCI-H82	Unknown

Figure viii. (a) RTK genes alterations in cancer cells; (b) The sensitivity of cancer cells to PHGDHi inhibitor. Cells were treated with CBR5884 at 6.25 μ M, 12.5 μ M or 25 μ M for 6 days. Heatmap depicts the inhibition rate of the cell growth.

5. Fig 2m, the co-amplification/deletion for *EGFR* and *PSPH* is likely caused by physical linkage of the two genes in chr 7p. Check other genes nearby may lead to the similar results. It does not prove that two genes have functional interactions.

Response: We thank the reviewer for the valuable comment. The reviewer is correct that the physical linkage of *EGFR* and *PSPH* should be considered. We have re-analyzed this data and corrected our description in the revised manuscript.

1) *PSPH* amplification status in Fig. 2m (Fig. 21 in the revised manuscript) was analyzed and annotated by cBioPortal public dataset through GISTIC algorithm, a widely-used method able to differentiate the focal alteration and chromosome level copy number gain^{3,4}. *PSPH* amplification described in this study is a focal amplification rather than chromosome level copy number gain.

2) As the reviewer suggested, we further analyzed the genetic alterations in 5 genes distributed within a 3-Mb region adjacent to *PSPH* in Chr 7p 11.2, which is within the range of focal alteration. We discovered the concurrent amplification of other genes in this focal region. There is a possibility that the co-occurrence of *PSPH* amplification and *EGFR* mutation could simply due to its physical linkage with other genes in this region. We have made corrections to remove any comments about their possible functional interactions.

Figure ix. The genetic alterations in genes located within 3-Mb region adjacent to *PSPH* gene in

Chr 7p 11.2. Genes were ranked as the sequence in chromosome. Data were based on 1940 NSCLC patients from TCGA database and visualized through cBioPortal.

6. Fig 3h: what were results from human cancer cell lines and clinical specimens, similar changes as in *Baf3* cells?

Response: The results discovered in BAF3 cells have been confirmed in clinical specimens by providing 3 sets of new results.

1) FGFR amplification associated upregulation of these enzymes were all confirmed in 740 lung adenocarcinoma patient samples (Fig. 3i).

Figure x. The comparison of glycolytic gene expression between FGFR amplified and diploid cancer from 740 lung adenocarcinoma patients in TCGA data sets.

2) We conducted IHC analysis in a panel of NSCLC PDX tumor samples. By comparing the NSCLC tumors with amplified (n=4) or wildtype (n=6) *FGFR*, we confirmed the increased expression of HK2 and LDHA in *FGFR*-amplified PDX samples. These results have been provided in the revised manuscript (Supplemental Fig. 3d) and the images of presentative models are also shown below.

Figure xi. Immunohistochemistry analysis of tumor tissues from representative NSCLC PDX tumors with wildtype or amplified *FGFR1/2*.

3) Furthermore, we attempted to get additional information from publicly available datasets. We used Project Achilles dataset that reveals the cell growth dependency on certain genes obtained by

CRISPR/Cas9 screening. As shown below, the dependency on LDHA, PFKL and PKM was significantly higher in *FGFR*-amplified cancer cells than that in *FGFR* wildtype cancer cells, while the dependency on PHGDH and PSPH showed no difference between the indicated subgroups, suggesting that *FGFR*-amplified cells were preferentially dependent on glycolysis. These results have been included in the revised manuscript (Supplemental Fig. 3e)

Figure xii. Dependency score for metabolic genes in cell lines with (Amp) or without (WT) *FGFR* amplification. Data were extracted from public dataset Project Achilles.

Reviewer #2 (Remarks to the Author): Expert in Cancer metabolism

Jin et al report a dissection of metabolic vulnerabilities associated with specific receptor tyrosine kinase (RTK) alterations encountered in tumors. They modeled different RTK effects in the context of a common cell line (BAF3). This analysis has led them to the identification of differential metabolic pathway utilization in the context of aberrant EGFR and FGFR. In the first case, the cells relied more heavily on the serine biosynthetic pathway. In the second case, the cells relied more heavily on glycolysis. The authors then extrapolate their findings beyond the BAF3 model to cancer cell lines endogenously harboring some of the oncogenic RTK alterations. Finally, authors elucidate the network of transcription factors controlling the differential metabolism observed. Collectively, this work aims to identify novel nodes for therapeutic intervention in specific cancer subsets.

While the authors choose to address an important question in the field of oncology and tumor metabolism, there are MULTIPLE SIGNIFICANT concerns that would have to be addressed before the manuscript would be considered ready for publication.

Response: We thank the reviewer for the comment that we “**choose to address an important in the field of oncology and tumor metabolism**”. We are also grateful for the multiple concerns raised by the reviewer. We have addressed these concerns by providing significant amount of new data or providing detailed explanations. We believe the reviewer’s valuable comments have helped us greatly improve the study.

Major comments:

- Metabolic changes in cancer cell lines are heavily dependent on cell culturing conditions. It is currently not clear from the manuscript to what extent some of the changes are driven by the media differences vs oncogenes encountered in the cell lines. The BAF3 experiments using isogenic cell lines are internally controlled. However, comparisons across different cancer cell lines should be done in the same media to enable direct comparisons.

[Redacted]

2) Of note, regardless of the various culture conditions, our results showed that cell lines carrying the same genetic alteration tended to show a similar metabolic dependency. Namely, EGFR cells are more dependent on serine synthesis whereas FGFR cells exhibit the increased dependency on lactate production (Fig. 2c, 3k). On the other hand, for quite a few *EGFR* or *FGFR* aberrant cells that were maintained in the same media, their growth dependency appears quite different.

Table i. Culture condition of FGFR and EGFR altered cells

Name	Gene alterations	Culture Media (plus FBS)
FGFR alteration		
NCI-H520	FGFR1 amplification	RPMI-1640+1.25 mM Pyruvate+2.5 g/L Glucose
NCI-H1581	FGFR1 amplification	DMEM/F12
NCI-H2444	FGFR1 amplification	RPMI-1640+1.25 mM Pyruvate+2.5 g/L Glucose
NCI-H716	FGFR2 amplification	RPMI-1640+1.25 mM Pyruvate+2.5 g/L Glucose
SNU-16	FGFR2 amplification	RPMI-1640
KATO III	FGFR2 amplification	IMDM
RT112	FGFR3 amplification, FGFR3-TACC3 fusion	RPMI-1640
UM-UC-14	FGFR3 S249C mutation	EMEM
EGFR alteration		
PC9	EGFR delE746-A750	RPMI-1640+1.25 mM Pyruvate+2.5 g/L Glucose
HCC827	EGFR delE746-A750	RPMI-1640+1.25 mM Pyruvate+2.5 g/L Glucose
HCC4006	EGFR delE746-A750	RPMI-1640
NCI-H1650	EGFR delE746-A750	RPMI-1640+1.25 mM Pyruvate+2.5 g/L Glucose
HCC2279	EGFR delE746-A750	RPMI-1640
NCI-H1975	EGFR L858R , T790M	RPMI-1640
NCI-H2172	EGFR L858R	RPMI-1640
NCI-H3255	EGFR L858R	BEGM

Figure ii. Cell sensitivity to PHGDH or LDH inhibition. Cells were treated with PHGDHi (CBR5884, 12.5 μ M) or LDHi (GSK 2837808A, 50 μ M) for 6 days. Scatter plot showing the inhibition rate on cell

lines with indicated genetic alteration. Wildtype cancer cell lines lacking detectable driving gene alterations were used as control.

- It's not clear whether the BAF3 experiments were done in the presence or absence of IL3. It is important to annotate this clearly. To what extent would the answer be different in the +/- IL3 conditions?

Response: We have clarified the experiment conditions and provided the news results, which largely eliminate the IL3's influence on the observed metabolic vulnerabilities.

1) All the experiments using BAF3-RTK cells were conducted in the absence of IL3, and hence largely excluded the impact of IL3 on the metabolic phenotypes. The parental BAF3 cells were cultured in the media containing IL3 due to their growth dependency. This information has been clarified in the revised manuscript.

2) Upon the reviewer's request, we have conducted new experiments to compare the metabolic phenotypes in the +/- IL3 conditions. As shown by the Seahorse XF96 analysis, deprivation of IL3 resulted in striking difference in BAF3 parental cells, which are expected as the survival of these cells are highly dependent on IL3. BAF3-RTK cells were generally less affected but with different extent (see the results below). This appeared also correlated with the impact of IL3 on cell growth.

Figure iii. The impact of IL3 on OCR and cell growth. BAF3-RTK or the parental BAF3 cells were culture in the presence/absence of IL3. (a, b) OCR were measured using Seahorse XF96 analyzer; (b) Growth curves were plotted by measuring growth fold change by counting cell numbers at indicated time.

- Figure 1b. The expression analysis of TCGA samples in superficial. Given that some of the tumor genotypes cluster by tissue type (e.g., EGFR mutations are encountered in lung cancer, but not in AML), it's not clear that metabolic clusters presented are not just a reflection of tissue types. Therefore, similar analysis of KEGG enzymes should be done within a particular cancer type, thus accounting for the confounding factor of tissue of origin. Any tumor genotype associations identified is such a way, would be significantly more meaningful.

Response: The reviewer raised a very important point. As suggested, we re-analyzed the TCGA 740 lung adenocarcinoma samples to eliminate the bias from different tissue type. We found that *EGFR* activation mutation, *FGFR* amplification and *RET* gene fusion are better clustered in metabolic gene expression pattern (Supplementary Fig. 1b, see also Figure iv below). Further looking into the KEGG pathways in these samples discovered that serine and related metabolism pathways were upregulated in *EGFR* mutated cancer patients (n = 25) versus the counterpart with wildtype *EGFR* (n = 715) (Fig. 2a).

Also, as suggested by the reviewer, for all the data related to analysis of TCGA samples (Fig. 2a, 2j, 3i, Supplementary Fig. 1b), we re-analyzed the data by specifically stratifying the lung adenocarcinoma samples, and the new results confirmed the metabolic phenotypes discovered in this study.

Figure iv. Heatmap obtained by cluster analysis using the mRNA data of metabolic genes of 740 lung adenocarcinoma patients in the TCGA data sets.

- The authors do a nice job of studying particular genetic lesions using the BAF3 model system. However, the follow up in endogenous cell lines panels is not sufficiently extensive. Therefore, the case for specificity of vulnerabilities associated with each of the genotypes is somewhat weak. One way to overcome this is to increase the number of cell lines analyzed. Alternatively, is there additional information that can be gleaned from publicly available datasets, particularly ones focusing on tumor dependence (Project Achilles, Project DRIVE, depmap, etc.)?

Response: We thank the reviewer for this important comment. Following the reviewer’s advice, we have provided 3 sets of new data, which further strengthened the metabolic vulnerability identified in *EGFR*- and *FGFR*- aberrant cancer.

1) We have increased the number of cell lines analyzed. Specifically, 8 *EGFR* mutant cancer cell lines, 8 *FGFR* amplified/mutant cancer cell lines and 8 wildtype lung cancer cell lines (except A431 which is an epidermoid carcinoma cell line often used as a control cell for RTK-related study) lacking genetic driving alterations, were selected to test their sensitivity to the inhibition of serine synthesis (PHGDH inhibitor) or lactate production (LDH inhibitor). Consistent with the results observed in BAF3 cell lines, *EGFR*-mutant cancer cells showed preferential sensitivity to PHGDH inhibition whereas *FGFR*-altered cancer cells were relatively more sensitive to LDH inhibitor. Wildtype cells, which are lack of driving alterations, showed heterogeneous outcomes.

Figure v. Cell sensitivity to PHGDH or LDH inhibition. Cells were treated with PHGDHi (CBR5884) at 6.25 μM , 12.5 μM and 25 μM , or LDHi (GSK 2837808A) at 12.5 μM , 25 μM and 50 μM for 6 days. (a) Scatter plot showing the inhibition rate on cell lines with indicated genetic alteration at 12.5 μM PHGDHi and 50 μM LDHi. Wildtype cancer cell lines lacking detectable driving gene alterations were used as control. (b) Heatmap depicts the inhibition rate of the cell growth.

2) We used Project Achilles dataset that screens the cell dependency on certain genes using CRISPR/Cas9 approach. As shown by Figure vi below, *FGFR*-amplified cancer cells showed significantly increased the dependency on LDHA, PFKL and PKM compared to the wildtype counterpart. As a good control, the dependency on PHGDH and PSPH showed no difference. The same approach could not be used for *EGFR*-mutated cancer cells, as the database only included 3 cell lines with altered *EGFR*.

Figure vi. Dependency score for metabolic genes in cell lines with (Amp) or without (WT) FGFR amplification. Data were extracted from public dataset Project Achilles.

3) We also took advantage of Cancer Cell Line Encyclopedia (CCLE) dataset to explore the metabolic signatures associated with EGFR/FGFR altered status. 91 lung cancer cell lines were divided into two groups according to the sensitivity to EGFR inhibitor (Erlotinib_pos or Erlotinib_neg defined according to Pearson correlation coefficient). Gene Set Enrichment Analysis (GSEA) analysis showed that genes in Erlotinib_pos group (EGFR dependent cells) were significantly enriched in pathway of GLYCINE_SERINE_AND_THREONINE_METABOLISM, consistent with the rest of the study.

Figure vii. Gene Set Enrichment Analysis (GSEA) analysis of two significantly enriched classes of genes: Erlotinib_pos or Erlotinib_neg according to the sensitivity to EGFR inhibitor (Erlotinib).

- The *in vivo* experiment shown in Figure 3d is potentially nice. However, it's hard to interpret without seeing the fractional enrichment in the blood, as only tumor results are shown.

Response: We thank the reviewer for the valuable comment. The fractional enrichment in the serum data have been provided (Supplementary Fig. 3b). According to this result, the serum lactate M3 and M1 fractions are quite similar, about 20%. This shows that the lactate from peripheral conversion of glucose or lactate from injection in the serum were equivalent. As the reviewer suggested, tumor fractions have been normalized to the serum lactate M3 and M1, respectively (Fig. 3d).

Figure viii. (a) Serum fractional enrichment of lactate. Mice with flank xenografts of H1581 cells were co-injected with [U-¹³C₆]-glucose and [3-¹³C]-lactate intravascularly and the serum were collected at for 30 min (n = 5 mice per group). (b) The incorporation percentage of [U-¹³C₆]-glucose and [3-¹³C]-lactate to TCA cycle. The ¹³C-enrichment was normalized to the lactate M3 and M1 in the serum, respectively.

- In Figure 3k the labeling results are potentially quite nice. However, showing citrate+2 results would significantly strengthen the authors' case. The choice of citrate+6 is peculiar, given the fact that it's likely to be found in exceedingly small amounts in the tumors and very close to the background noise.

Response: We thank the reviewer for pointing this out. Fig 3k (Fig. 3l in the revised manuscript) was mislabeled. It was the isotopologue citrate M2 rather than citrate M6. In fact, the isotopologue citrate M6 cannot be detected due to the short time of infusion. We apologized for the mistake and have made corrections in the revised manuscript.

Figure ix. (a) Tumor growth curve of SNU16 xenograft. (b) Analysis of glucose-derived metabolites in tumor tissues. Mice bearing SNU16 tumors were subjected to bolus injection of [U-¹³C₆]-glucose tracer.

- While oxamate is a reasonable starting tool compound for inhibiting lactate dehydrogenase, it lacks specificity (PMID: 27538376). Therefore, at least in vitro, some additional inhibitors should be used to support authors' claims. Fortunately, multiple specific lactate dehydrogenase inhibitors have been published to date (PMID: 27479743, 22417091, 23302067, 24280423)

Response: We appreciate the reviewer's comments. As suggested by the reviewer, we included another more selective LDH inhibitor, GSK2837808A in the study to strengthen our findings. The new results have been included in the revised manuscript and also attached below.

1) We have shown that GSK2837808A inhibition similarly decreased OCR (Fig. 3e), ATP production (Fig. 3f) and ROS production (Fig. 3g) like oxamate in *FGFR* amplified cells.

Figure x. (a) OCR measurement. NCI-H1581 cells were treated with AZD4547 (100 nM, 24 hr) , Oxamate (10 mM, 6 hr) or GSK2837808A (20 μ M, 6 hr). (b) ATP production and ROS level. NCI-H1581 cells were treated as indicated in a.

2) We also showed that in a cancer cell panel composed of 8 *EGFR* mutant cancer cell lines, 8 *FGFR* amplified/mutant cancer cell lines and 8 wildtype cancer cell lines, *FGFR* cells are relatively more sensitive to GSK2837808A.

a

Name	Gene alterations	Name	Gene alterations	Name	Gene alterations
NCI-H520	FGFR1 amplification	PC9	EGFR delE746-A750	A431	Unknown
NCI-H1581	FGFR1 amplification	HCC827	EGFR delE746-A750	NCI-H1299	Unknown
NCI-H2444	FGFR1 amplification	HCC4006	EGFR delE746-A750	NCI-H2170	Unknown
NCI-H716	FGFR2 amplification	NCI-H1650	EGFR delE746-A750	ABC-1	Unknown
SNU-16	FGFR2 amplification	HCC2279	EGFR delE746-A750	ChaGo-K-1	Unknown
KATO III	FGFR2 amplification	NCI-H1975	EGFR L858R , T790M	NCI-H226	Unknown
RT112	FGFR3 amplification, FGFR3-TACC3 fusion	NCI-H2172	EGFR L858R	SK-MES-1	Unknown
UM-UC-14	FGFR3 S249C mutation	NCI-H3255	EGFR L858R	NCI-H82	Unknown

Figure xi. (a) RTK genes alterations in cancer cells; **(b)** Cell sensitivity to PHGDH or LDH inhibition. Cells were treated with PHGDHi (CBR5884) at 6.25 μ M, 12.5 μ M and 25 μ M, or LDHi (GSK 2837808A) at 12.5 μ M, 25 μ M and 50 μ M for 6 days. Left, Scatter plot showing the inhibition rate on cell lines with indicated genetic alteration at 12.5 μ M PHGDHi and 50 μ M LDHi. Wildtype cancer cell lines lacking detectable driving gene alterations were used as control. Right, Heatmap depicts the inhibition rate of the cell growth.

- The connection of ATF4 transcription factor regulating the serine biosynthetic pathway is not novel (PMID: 26482881) and the authors don't cite it. Furthermore, authors need to account for the NRF2 pathway connection (via ATF4 activation) to the serine pathway. This suggestion pertains to bioinformatic analyses (accounting for EGFR mutations vs NRF2 pathway activation mutations) and to the choice of cell lines used for follow-up studies.

Response: We were aware of this important work from Cantley's laboratory and have cited it in the revised manuscript. We have also provided new results to dissect role of NRF2 in EGFR mutant cancer, and discussed the difference between Cantley's work and ours.

1) As suggested by the reviewer, we have provided new results, which showed that in EGFR mutant cells, knockdown of NRF2 downregulates ATF4 expression as well as downstream metabolic enzymes (Supplementary Fig. 4d, see also Figure xii below). This data, together with literatures^{5,6}, have positioned NRF2 upstream of ATF4 in EGFR mutant context in NSCLC.

Figure xii. Knocking down NRF2 downregulates ATF4 and metabolic enzymes in EGFR mutant NSCLC. PC9 cells were transfected indicated siRNA for 72 hr and the expression of indicated genes was analyzed by qPCR.

2) The key difference between Cantley's work and this study is that we focused the different subsets of NSCLC. Cantley's work was inspired by the frequent occurrence of NRF2 activation mutation in NSCLC. Interestingly, NRF2 mutation occurs often in NSCLC but appears very rare in EGFR mutant cancer, which is the focus of our study. According to TCGA database, among 1940 lung

carcinoma samples (including 44 *EGFR* mutant patients), 106 patients showed *NRF2* mutation yet none of them co-occurred with *EGFR* mutation. Our work may provide a complimentary mechanism for upregulating the serine biosynthetic pathway via *NRF2-ATF4* axis in lung cancer.

Minor comments:

- *Figure 1c. The axis is labeled “proliferation curve (relative)”. A more accurate description should be something along the lines of “relative cell number”. Similar comment applies to all figures displaying relative rates of cell growth.*

Response: We have made the corrections in all related figures following the reviewer’s advice.

- *There are several figure panels that are completely not discussed in the text of the manuscript (Figure 2d, Supplemental Figure 2f, etc.). It’s worth adding some text to describe them.*

Response: Thank the reviewer for the comment. We have gone through the manuscript to ensure that all the figures, including supplemental figures, are discussed in the text of the revised manuscript.

- *The authors are very strong in some of their conclusions, even in the context of sparse data. It’s strongly recommended that they adopt a more conservative language to state their potential conclusions. For example, line 126 (“...faithfully echoed...”). For example, line 137 (the text states that CB839 only affected *RET*-aberrant cells). Certainly, *RET* cells were the most impacted, but other cell lines showed significant effects as well. For example, line 261 (“... proving our findings in vivo.”).*

Response: We thank the reviewer for the important comment. All these sentences have been rewritten to avoid overstatement.

- *Only one of the endogenous cell lines harboring *RET* alteration displayed differential growth upon glutamine withdrawal (TT), while the other cell line (TPC-1) was not particularly affected. This data is shown in Supplemental Figure 1i, but the text in the manuscript overstates the case by saying that all *RET* lines were affected.*

Response: We thank the reviewer for pointing this out. We could not test more *RET* altered cells due to the limited resources of *RET* aberrant cell lines. As the reviewer suggested, we have adopted a more conservative language to avoid the overstatement for this result, which are stated as “*RET*

fusion cells seemed to rely on glutamine for proliferation, which yet remained to be further confirmed due to the very limited number of RET-aberrant cancer cells available for this study”.

- Figure 1i. Metabolic pathway enrichment is shown here as function of the BAF3 genotype. However, it would also useful to see all pathways that are enriched, and not restricting the analysis to metabolism only. This will give a reader a better idea whether metabolic rewiring is a major component of what is happening downstream of these RTKs or not.

Response: We thank the reviewer for the advice and have provided a complete list of all the significant KEGG pathways that are enriched according to distinguished clusters (Supplementary Table 6).

- Supplementary Figure 2a is not discussed in the text. The labels in the left panels are likely incorrect. Presumably, they are meant to compare BAF-EGFR vs BAF-FGFR.

Response: The reviewer is correct. The figure was mislabeled and should be a comparison between BAF3-EGFR vs BAF3-FGFR. We apologize for the mistake. We have made the correction and discussed the figure in the revised manuscript (Supplementary Fig. 2a in the revised manuscript).

- The manuscript text overstates the case of specificity of PHGDH and PSPH effects on EGFR-aberrant cell lines. Supplementary Figure 2g clearly shows that 1 out of the 2 FGFR-aberrant cell lines also is affected by PHGDH and PSPH knockdown. Does the left most panel in that figure contain a correct label or correct control? Right now “EGFR” is used as a control here, but FGFR would make more sense as a control. The left most panel in Fig. 2g should be FGFR1 as a control.

Response: We thank the reviewer for this important comment. The questions are addressed point-to-point below.

1) To strengthen the observed specificity of PHGDH or PSPH effect on EGFR-aberrant cancer, we have expanded the cancer cell lines (8 *EGFR* mutant cancer cell lines, using 8 *FGFR* amplified/mutant and 8 wildtype cancer cell lines as control cells) to test their dependency on a PHGDH inhibitor. As shown below, *EGFR* mutant cells are generally more sensitive to the PHGDH inhibition compared with *FGFR* amplified cells. Wildtype cells, which lack driving gene alterations, are more heterogeneous and showed variable phenotypes as expected, and most of them are not responsive to the treatment. This result has been included in the revised manuscript (Fig. 2c) and also shown below.

Figure xiii. The sensitivity of cancer cells to PHGDH inhibitor. Cells were treated by CBR5884 at 6.25 μ M, 12.5 μ M or 25 μ M for 6 days. Heatmap depicts the inhibition rate of the cell growth.

2) The reviewer is correct, Supplementary Fig. 2g (Supplementary Fig. 2i in the revised manuscript) was mislabeled. It should be FGFR1 for the left panel. We have made the corrections.

Figure xiv. Cell viability change upon knockdown of indicated genes in FGFR1/2 amplified aberrant cancer cells.

- PHGDH inhibitor used, CBR5584, is not cited in the bibliography.

Response: The reference (*Proc Natl Acad Sci USA*. 2016, 113:1778-83) has been added in the revised manuscript.

- Line 198. The discussion of PSPH being a rate-limiting enzyme of the serine biosynthetic pathway is incomplete. While this enzyme can be the rate-limiting in the context of the liver, it may not be in the context of cancer. Please see PMID: 21760589.

Response: Thanks for reviewer's comment. We have made the correction and cited the study (*Nature*. 2011,476: 346-50) as suggested.

- The relevance of the findings shown in Figure 2l is not particularly clear. Similarly, the analysis

shown in Figure 2m is not clearly described. What are the cutoffs used for defining PSPH amplified tumors? Is it a focal amplification, similar to the one encountered with PHGDH? Alternatively, is it a broad chromosome-wide copy number gain (in which case it would be harder to make a case for PSPH contribution)? Please provide some statistical analysis that these two co-occurring events (EGFR mutation and PSPH amplification) are significant, beyond what would be expected by chance.

Response: We thank the reviewer for the questions. We have re-analyzed this data as suggested and made changes accordingly.

1) Figure 2l in the previous manuscript intended to show the functional interaction between EGFR and PSPH, in which patients were divided into two groups according to the PSPH expression using the median of PSPH expression as a cutoff. We agree with the reviewer that the relevance of the findings in this figure is low and hence removed this data in the revised manuscript.

2) PSPH amplification status in Fig. 2m (Fig. 2l in the revised manuscript) was analyzed and annotated by cBioPortal public dataset through GISTIC algorithm, a widely-used method able to differentiate the focal alteration and chromosome level copy number gain^{3,4}. PSPH amplification described in this study is a focal amplification rather than chromosome level copy number gain.

3) Responding to reviewer's question, we have compared the alteration of PSPH and PHGDH in parallel. Within a 3-Mb region, which is within the range of focal alteration, we discovered the associated amplification of adjacent genes for both PSPH and PHGDH. In contrast, within a 10 Mb region, the incidence of associated gene amplification was largely reduced for both cases (Figure xv below). These results show that both PSPH and PHGDH are focal amplifications.

Figure xv. Upper, The alteration of indicated genes adjacent to PHGDH in Chr 1p. Lower, The alteration of indicated genes adjacent to PSPH in Chr 7p 11.2. Genes were ranked as the sequence in chromosome. CD53, EMBP1, ZNF733P and VWC2 represented genes distributed in 10-Mb region and the rest are within 3 Mb. Data were from NSCLC patients in TCGA and visualized through cBioPortal.

4) As the reviewer suggested, we have conducted statistical analysis using Fisher's exact test and discovered the significant co-occurrence of *EGFR* mutation and *PSPH* amplification in patients samples ($p = 0.0067$).

- Figure 3c label "Lactate relative absolute intensity" is not clear. Please explain.

Response: We apologize for unclear labeling in the figure. It intended to show the metabolite enrichment, as calculated by the peak area of the isotopologue lactate M3 per 10^7 cells that was normalized by the peak area of internal standard.

We have revised the labeling to avoid confusion and the figure is also shown below. The ^{13}C -enrichment of the metabolites derived from $[\text{U-}^{13}\text{C}_3]$ -lactate at the steady state (24 hr) in FGFR1 and EGFR-driven cells were presented.

Figure xvi. Fraction contribution of the metabolites in BAF3-FGFR1 and -EGFR cells generated from $[\text{U-}^{13}\text{C}_3]$ -lactate.

- Please provide more details on how the amplifications were defined in Figure 3i.

Response: *FGFR1* gene amplification was annotated by the cBioPortal public dataset through GISTIC algorithm. This information has been added in the revised manuscript.

- Line 269. Instead of the phrase "...mitochondrial phosphorylation..." is should read "...mitochondrial respiration...".

Response: Thank the reviewer for the comment. We have made the correction as suggested.

References

1. Siolas D, Hannon GJ. Patient-derived tumor xenografts: transforming clinical samples into mouse models. *Cancer Res* **73**, 5315-5319 (2013).
2. Tjin Tham Sjin R, *et al.* In vitro and in vivo characterization of irreversible mutant-selective EGFR inhibitors that are wild-type sparing. *Mol Cancer Ther* **13**, 1468-1479 (2014).
3. Mermel CH, Schumacher SE, Hill B, Meyerson ML, Beroukhim R, Getz G. GISTIC2.0 facilitates sensitive and confident localization of the targets of focal somatic copy-number alteration in human cancers. *Genome Biol* **12**, R41 (2011).
4. Beroukhim R, *et al.* Assessing the significance of chromosomal aberrations in cancer: methodology and application to glioma. *Proc Natl Acad Sci U S A* **104**, 20007-20012 (2007).
5. Yamadori T, *et al.* Molecular mechanisms for the regulation of Nrf2-mediated cell proliferation in non-small-cell lung cancers. *Oncogene* **31**, 4768-4777 (2012).
6. DeNicola GM, *et al.* NRF2 regulates serine biosynthesis in non-small cell lung cancer. *Nat Genet* **47**, 1475-1481 (2015).

Reviewers' Comments:

Reviewer #1:

Remarks to the Author:

The authors addressed my previous comments. Here are some minor comments:

1. Figure 1j, what is the legend for the first panel of bars?
2. Figure 2C, it will be helpful to provide IC50 etc instead of heatmap for sensitivity of an agent.

Reviewer #2:

Remarks to the Author:

The authors have very significantly improved the manuscript. There are several additional (but relatively minor) issues that need to be fixed. Pending their successful completion, the manuscript should be acceptable for publication.

Specific comments (science):

- Figure 2i. Include the results from BAF-FGFR1 cells as additional controls.
- Figure 3h. include the results from BAF-EGFR cells as additional controls.
- Provide evidence that the KEGG gene sets identified in TCGA as correlated with various RTK mutations (Figure S1b) overlap with the gene sets identified in the BAF3 model system used (Figure S1o). Without this direct evidence the conclusion sentence (lines 135-138) is overly strong.
- IL3 presence. The IL3 parameter is very important for the appropriate use of the BAF3 system. The authors have provided a satisfactory answer in their response. However, please incorporate this information into the manuscript. IL3 presence or absence should be indicated in the figure legends. Figure iii a and Figure iii b (from the response to reviewer letter) should be incorporated in supplementary data.
- The GSEA results shown in Figure vii appear to be nice, but the p values were not indicated. Given the fact that these results provide orthogonal support for the main conclusion, it might be worth considering their inclusion in the supplementary data.
- Functional genomics data displayed in Figure S3e is not adequate. The scale appears to be peculiar. Standard way of displaying this data shows dependency with negative values, zero indicates no effect, and positive values indicate growth enhancement. No reference is given for the dataset used. No reference is given for the algorithm used to analyze the CRISPR data (presumably, CERES).
- Line 268. Physiological concentration of glucose in approximately 5mM. Please adjust the text accordingly.
- Lines 248-249. The lines make a statement about potential difference in survival of patients with EGFR mutations and PSPH amplification. No data is provided to support this claim. Either remove the claim or provide convincing evidence.
- Lines 254-255. The conclusion of this paragraph is overly strong in stating the PSPH might be a good cancer target in this context. The data to support this is relatively weak:
 - o EGFR aberrant cell lines: nice data in PC9 cells, moderate data in H1975.
 - o FGFR aberrant cell lines: nice data in DMS114, moderate data in RT112.
 - o If fact, the results in H1975 and RT112 cells are not really distinguishable. Either provide experiments in additional cell lines or significantly "soften" the conclusion.

Specific comments (manuscript assembly):

- Line 38. Current phrase "...FGFR addition..." is not fully accurate and should read "...FGFR activation...".
- Line 273. The sentence refers to Fig.2 d. Instead, it should be referring to Fig. 3d.
- Now the manuscript is almost ready for publication, it would be worth employing professional editing services to correct some of the language/editing issues. There are multiple examples of awkward phrases (e.g., lines 202, 206, 339, 412, 414-415, 436-437, etc.)

- Figure 1. Letter "h" is missing to label the panel to the right of panel g.
- Figure 2k. Describe when the tumors were harvested in the legend of the figure (end of the in vivo efficacy study?).
- There is additional data shown and discussed in the context of the Discussion section (Figure S4d). This is a bit peculiar. Consider displaying all primary data in the context of the Results section.
- Figure S3g. No data is shown for the AZD4547 arm of the experiment. Some data is shown for an experimental arm called "combo". Not clear what combo means in this context.

Point-to-point responses to reviewers' comments

Reviewer #1 (Remarks to the Author):

The authors addressed my previous comments. Here are some minor comments:

1. Figure 1j, what is the legend for the first panel of bars?

Response: FGFR1-activated cells showed two differentially transcribed clusters of metabolic genes in Supplementary Fig. 1p (left panel below). The first two panels in Figure 1j (right panel below) showed enriched metabolism-related KEGG pathways ($p < 0.05$) of these two clusters of genes. We thank the reviewer for reminding us. The missing label for the first panel in Figure 1j has been added.

Figure i. Transcriptome analysis of BAF3-RTK cells. Left panel: Heatmap of transcriptome profiling representing of the mRNA levels of genes performed by RNAseq. Right panel: KEGG pathway enrichment analysis by the differentially transcribed clusters among different cells. The different clusters were color-coded.

2. Figure 2C, it will be helpful to provide IC50 etc instead of heatmap for sensitivity of an agent.

Response: Following the reviewer's advice, we have provided a graph of the proliferation inhibition rate of each cell line at three different concentrations of PHGDH inhibitor (CBR5884) in Supplementary Fig. 2b (see also Figure ii below).

Figure ii. Sensitivity of a panel of cancer cells to PHGDH inhibition. Cancer cells with indicated genetic alterations were treated with CBR5884 at 6.25, 12.5 or 25 μ M for 6 days and the inhibition rate of cell growth was determined relative to untreated control.

Reviewer #2 (Remarks to the Author):

The authors have very significantly improved the manuscript. There are several additional (but relatively minor) issues that need to be fixed. Pending their successful completion, the manuscript should be acceptable for publication.

Specific comments (science):

1. *Figure 2i. Include the results from BAF-FGFR1 cells as additional controls.*

Response: We thank the reviewer for the advice and have included BAF3-FGFR1 cells as additional controls in the revised Figure 2i, line 221-225 (see also Figure iii below).

Figure iii. Relative expression level of indicated genes that normalized by that in BAF3 cells.

2. *Figure 3h. include the results from BAF-EGFR cells as additional controls.*

Response: As the reviewer suggested, we have included BAF3-EGFR cells as additional controls in the revised Figure 3h, line 285-289 (see also Figure iv below).

Figure iv. Relative expression level of indicated genes that normalized by that in BAF3 cells.

3. Provide evidence that the KEGG gene sets identified in TCGA as correlated with various RTK mutations (Figure S1b) overlap with the gene sets identified in the BAF3 model system used (Figure S1o). Without this direct evidence the conclusion sentence (lines 135-138) is overly strong.

Response: We thank the reviewer for this important comment. We have removed the strong conclusion sentences (line 134).

Also, as suggested, we analyzed the metabolic genes in EGFR- and FGFR-activated tumors displayed in Supplementary Fig. 1b. KEGG pathway enrichment analysis of these altered metabolic genes (1.5-fold cutoff; $p < 0.01$) highlighted several metabolic pathways in EGFR- and FGFR-activated tumors respectively, such as pyruvate metabolism in *FGFR* amplified tumors and glycine serine and threonine metabolism in *EGFR* mutant tumor, which overlapped with gene sets identified in the BAF3 cells (line 164-171). These results have been included in Supplementary Fig. 1q (see also Figure v below).

Figure v. KEGG pathway enrichment analysis the metabolic genes between EGFR- and FGFR-activated tumors that displayed in Supplementary Fig. 1b. The significantly enriched metabolism-

related KEGG pathways ($p < 0.05$) were presented. Bars show the enrichment score of the pathway and were presented according to p value.

4. IL3 presence. The IL3 parameter is very important for the appropriate use of the BAF3 system. The authors have provided a satisfactory answer in their response. However, please incorporate this information into the manuscript. IL3 presence or absence should be indicated in the figure legends. Figure iii a and Figure iii b (from the response to reviewer letter) should be incorporated in supplementary data.

Response: As the reviewer suggested, IL3 presence or absence have been indicated in all the related figure legends. The IL3 data have also been incorporated in Supplementary Fig. 1g (Line 108-113).

5. The GSEA results shown in Figure vii appear to be nice, but the p values were not indicated. Given the fact that these results provide orthogonal support for the main conclusion, it might be worth considering their inclusion in the supplementary data.

Response: GSEA results in Figure vii in the previous response letter suggested that Erlotinib_pos group showed a trend of pathway enrichment in serine-related metabolism according to high enrichment score. Reminded by the reviewer, we also looked into p values and they seemed not significant. We hence did not incorporate this data in the manuscript.

6. Functional genomics data displayed in Figure S3e is not adequate. The scale appears to be peculiar. Standard way of displaying this data shows dependency with negative values, zero indicates no effect, and positive values indicate growth enhancement. No reference is given for the dataset used. No reference is given for the algorithm used to analyze the CRISPR data (presumably, CERES).

Response: We have re-plotted this functional genomics data in Supplementary Fig. 3e in the way the reviewer suggested¹. Consistent with shown before, *FGFR*-amplified cancer cells showed significantly increased dependence on LDHA, PFKL and PKM compared to the wildtype tumors. As a good control, the dependence on serine synthesis genes (PHGDH and PSPH) did not show any difference.

The reviewer is correct. The algorithm we used to analyze the CRISPR data is CERES². This information and the references have been added in the revised manuscript (Line 295).

Figure vi. Dependency score for metabolic genes in cell lines with (Amp) or without (WT) FGFR amplification. Data were extracted from public dataset Project Achilles.

7. Line 268. *Physiological concentration of glucose in approximately 5mM. Please adjust the text accordingly.*

Response: We thank the reviewer for pointing this out and have corrected the sentence (Line 262-263).

8. Lines 248-249. *The lines make a statement about potential difference in survival of patients with EGFR mutations and PSPH amplification. No data is provided to support this claim. Either remove the claim or provide convincing evidence.*

Response: We thank the reviewer for pointing this out, and have removed this claim (Line 244).

9. Lines 254-255. *The conclusion of this paragraph is overly strong in stating the PSPH might be a good cancer target in this context. The data to support this is relatively weak:*

- o EGFR aberrant cell lines: nice data in PC9 cells, moderate data in H1975.*
- o FGFR aberrant cell lines: nice data in DMS114, moderate data in RT112.*
- o If fact, the results in H1975 and RT112 cells are not really distinguishable. Either provide experiments in additional cell lines or significantly “soften” the conclusion.*

Response: We agree that the difference in EGFR- and FGFR- aberrant cell lines was relatively mild due to the limited cell numbers tested in each genotype. We have modified the sentence to avoid being overstated (Line 245-248).

Specific comments (manuscript assembly):

1. Line 38. *Current phase “...FGFR addition...” is not fully accurate and should read “...FGFR activation...”.*

Response: We have made the correction in the revised manuscript (Line 35 and 377).

2. *Line 273. The sentence refers to Fig.2 d. Instead, it should be referring to Fig. 3d.*

Response: We thank the reviewer for identifying this mistake and have made the correction in the revised manuscript (Line 268).

3. *Now the manuscript is almost ready for publication, it would be worth employing professional editing services to correct some of the language/editing issues. There are multiple examples of awkward phrases (e.g., lines 202, 206, 339, 412, 414-415, 436-437, etc.)*

Response: Thank the reviewer for the comment. We have gone through the manuscript with the help of professional editing to avoid awkward phrases.

4. *Figure 1. Letter “h” is missing to label the panel to the right of panel g.*

Response: We have corrected the mistake and confirmed all the labeling in the revised manuscript.

5. *Figure 2k. Describe when the tumors were harvested in the legend of the figure (end of the in vivo efficacy study?).*

Response: Immunohistochemistry analysis of tumor tissues from LU-01-0251 PDX were collected at 6 hr after the last dosing. This information has been added in the figure legend (Line 804).

6. *There is additional data shown and discussed in the context of the Discussion section (Figure S4d). This is a bit peculiar. Consider displaying all primary data in the context of the Results section.*

Response: We thank the reviewer for the advice. Considering the literatures^{3,4} that have positioned NRF2 on the upstream of ATF4 in *EGFR* mutant NSCLC, we removed this data in the revised manuscript and claimed the as “data not shown” (Line 441).

7. *Figure S3g. No data is shown for the AZD4547 arm of the experiment. Some data is shown for an experimental arm called “combo”. Not clear was combo means in this context.*

Response: We thank the reviewer for pointing this out and have made corrections in the revised manuscript.

References

1. Cheung HW, *et al.* Systematic investigation of genetic vulnerabilities across

- cancer cell lines reveals lineage-specific dependencies in ovarian cancer. *Proc Natl Acad Sci U S A* **108**, 12372-12377 (2011).
2. Meyers RM, *et al.* Computational correction of copy number effect improves specificity of CRISPR-Cas9 essentiality screens in cancer cells. *Nat Genet* **49**, 1779-1784 (2017).
 3. Yamadori T, *et al.* Molecular mechanisms for the regulation of Nrf2-mediated cell proliferation in non-small-cell lung cancers. *Oncogene* **31**, 4768-4777 (2012).
 4. DeNicola GM, *et al.* NRF2 regulates serine biosynthesis in non-small cell lung cancer. *Nat Genet* **47**, 1475-1481 (2015).

Reviewers' Comments:

Reviewer #1:

Remarks to the Author:

The authors addressed my comments. The manuscript is publishable

Reviewer #2:

Remarks to the Author:

The authors addressed my previous comments.

One remaining small scientific request:

- Please indicate the statistical significance cut-off used in Figure S3e.

One remaining editing issue:

Despite authors' assurances, the manuscript has not been edited for English language at a sufficient editorial level or language proficiency. There are multiple examples of grammatically incorrect sentences that still remain (e.g., lines 402, 403-405, 417-418, etc.).

Point-to-point responses to reviewers' comments

Reviewer #1

The authors addressed my comments. The manuscript is publishable

Response: We thank the reviewer for the comment that the manuscript is publishable.

Reviewer #2

The authors addressed my previous comments.

Response: We are pleased that we have addressed the comments raised by the reviewer.

One remaining small scientific request:

- Please indicate the statistical significance cut-off used in Figure S3e.

Response: $p < 0.05$ was considered to be statistically significant in Figure S3e. This information has been included in revised manuscript.

One remaining editing issue:

Despite authors' assurances, the manuscript has not been edited for English language at a sufficient editorial level or language proficiency. There are multiple examples of grammatically incorrect sentences that still remain (e.g., lines 402, 403-405, 417-418, etc.).

Response: We thank the reviewer for bringing our attention to these language issues. We have edited the language in the sentences the reviewer pointed out, and have requested the professional assistance in the language editing for the manuscript.